# Genomes of multicellular algal sisters to land plants illuminate signaling network evolution

Xuehuan Feng[1,34], Jinfang Zheng [1,32,34], Iker Irisarri[2,3,4,34], Huihui Yu[5,33], Bo Zheng[1], Zahin Ali[6], Sophie de Vries [2], Jean Keller[7], Janine M. R. Fürst-Jansen [2], Armin Dadras [2], Jaccoline M. S. Zegers[2], Tim P. Rieseberg [2], Amra Dhabalia Ashok [2], Tatyana Darienko [2], Maaike J. Bierenbroodspot [2], Lydia Gramzow [8], Romy Petroll [9,10], Fabian B. Haas [9,10], Noe Fernandez-Pozo [9,11], Orestis Nousias[1], Tang Li[1], Elisabeth Fitzek [12], W. Scott Grayburn[13], Nina Rittmeier[14], Charlotte Permann[14], Florian Rümpler[8], John M. Archibald[15], Günter Theißen[8], Jeffrey P. Mower [5], Maike Lorenz [16], Henrik Buschmann[17], Klaus von Schwartzenberg[18], Lori Boston[19], Richard D. Hayes [20], Chris Daum [20], Kerrie Barry [20], Igor V. Grigoriev [20,21,22], Xiyin Wang[23], Fay-Wei Li [24,25], Stefan A. Rensing [9,26], Julius Ben Ari[27], Noa Keren [27], Assaf Mosquna [27], Andreas Holzinger [14], Pierre-Marc Delaux [7], Chi Zhang[5,28], Jinling Huang [29,30], Marek Mutwil[6], Jan de Vries [2,3,31,34] ✉ & Yanbin Yin [1,34] ✉

Zygnematophyceae are the algal sisters of land plants. Here we sequenced four genomes of filamentous Zygnematophyceae, including chromosome-scale assemblies for three strains of *Zygnema circumcarinatum*. We inferred traits in the ancestor of Zygnematophyceae and land plants that might have ushered in the conquest of land by plants: expanded genes for signaling cascades, environmental response, and multicellular growth. Zygnematophyceae and land plants share all the major enzymes for cell wall synthesis and remodifications, and gene gains shaped this toolkit. Co-expression network analyses uncover gene cohorts that unite environmental signaling with multicellular developmental programs. Our data shed light on a molecular chassis that balances environmental response and growth modulation across more than 600 million years of streptophyte evolution.

Plant terrestrialization changed the surface of the Earth. The first land plants (Embryophyta) emerged from within the clade of Streptophyta about 550 million years ago[1]. Among six classes of streptophyte algae, the closest relatives of land plants are the Zygnematophyceae[2–4], algae with more than 4,000 described species[5] arranged into five orders[6]. So far, genome sequences are available only for unicellular Zygnematophyceae[7–9].

Zygnematophyceae possess adaptations to withstand terrestrial stressors, such as desiccation, ultraviolet light, freezing and other abiotic stresses[10]. The nature of these stress responses is of deep biological importance: various orthologous groups of proteins once considered specific to land plants have recently been inferred to predate the origin of Embryophyta[11,12]. The accuracy of inferring the developmental and

**Fig. 1 | Zygnema. a**, Three cells of a vegetative filament of SAG 698-1b (top) compared with one cell of a vegetative filament of SAG 698-1a (bottom, both samples of 1 month old cultures). Scale bar, 20 µm. C, chloroplast; N, nucleus; P, pyrenoid. One-cell filament contains two chloroplasts and one nucleus. **b**, Chromosome counting on light micrographs of SAG 698-1b fixed and stained with acetocarmine at prophase (0.5 months old); count was also performed in metaphase and telophase (Supplementary Fig. 1). The green dots represent the 20 chromosomes that were counted after rendering a stack of ~100 images. Scale bar, 10 µm. See Supplementary Fig. 1 for the original images. A minimum of ten cells each from three independent cell cultures were analyzed. **c**, A confocal laser scanning image of one SAG 698-1b cell (0.5 months). Scale bar, 20 µm. **d**, Transmission electron micrographs illustrating the filamentous nature of *Z. circumcarinatum* (SAG 698-1b). Left: overview showing that the cells are connected by extremely thin cross cell walls (cCW), while the outer cell wall (CW) is surrounded by a pectinous extracellular matrix (ECM); within the individual

cells, pyrenoids (Py) and the nucleus (N) are clearly depictable. Scale bar, 5 µm. **e**, A detailed view of the cross wall separating two cells where chloroplast lobes are visible. Scale bar, 0.5 µm. Transmission electron micrographs (**d** and **e**) derived from the analysis of ≥15 algal filaments each for two independent cell cultures. **f**, Chromosome-level assembly of the SAG 698-1b genome. Concentric rings show chromosome (Chr) numbers, gene density (blue), repeat density (yellow), RNA-seq mapping density $\log_{10}$(fragments per kilobase of transcript per million mapped reads) (dark green) and guanine-cytosine content density (violet). The red and green links show respectively intra- and interchromosomal syntenic blocks. **g**, A comparison of genome properties for 13 algal and 3 land plant species. The time-calibrated species tree was built from 493 low-copy genes (all nodes supported by >97% nonparametric bootstrap; numbers at branches are estimated divergence times in million years (mean ± standard deviation) (see Supplementary Table 1f for details). Data for the bar plot can be found in Supplementary Table 1i,j.

physiological programs of the first land plant ancestors depends on our ability to predict them in its sister group.

In this Article, we report on the first four genomes of filamentous Zygnematophyceae, including the first chromosome-scale assemblies for any streptophyte algae. By using comparative genomics, we

pinpoint genetic innovations of the earliest land plants. Our network analyses reveal co-expression of genes that were expanded and gained in the last common ancestor (LCA) of land plants and Zygnematophyceae. We shed light on the deep evolutionary roots of the mechanism for balancing environmental responses and multicellular growth.

**Table 1 | Genome assembly statistics for the new *Zygnema* genomes and available streptophyte algae (see Supplementary Table 1b–e for further details)**

| Species (strain) | Assembly size (Mb) | BUSCO (%) | N50 (kb) | Number of scaffolds (pseudochromosomes) | RNA-seq mapping rate (%) |
|---|---|---|---|---|---|
| *Z. circumcarinatum* SAG 698-1b | 71.0 | 89.8 | 3,958.3 | 90 (20) | 97.2 |
| *Z. circumcarinatum* UTEX 1559 | 71.3 | 88.2 | 3,970.3 | 614 (20) | 98.3 |
| *Z. circumcarinatum* UTEX 1560 | 67.3 | 87.9 | 3,792.7 | 514 (20) | 95.9** |
| *Z.* cf. *cylindricum* SAG 698-1a_XF | 359.8 | 70.6 | 213.9 | 3,587 | 88.3 |
| *Mesotaenium endlicherianum* SAG 12.97 | 163 | 78.1 | 448.4 | 13,861 | 94.4 |
| *Penium margaritaceum* SAG 2640 | 3,661 | 49.8 | 116.2 | 332,786 | 96.8 |
| *Spirogloea muscicola* CCAC 0214 | 174 | 84.7 | 566.4 | 17,449 | 95.2 |
| *Chara braunii* S276 | 1,430 | 78.0 | 2,300 | 11,654 | 89.5 |
| *Klebsormidium nitens* NIES-2285 | 104 | 94.9 | 134.9 | 1,814 | 98.1 |
| *Chlorokybus melkonianii* CCAC 0220 | 74 | 93.3 | 752.4 | 3,809 | 96.5 |
| *Mesostigma viride* CCAC 1140 | 281 | 59.2 | 113.2 | 6,924 | 84.3 |

**The mapping rate of *Z. circumcarinatum* UTEX 1560 was calculated by using SAG 698-1b RNA-seq reads mapped to the genome of UTEX 1560.

## Results

### First chromosome-level genomes for streptophyte algae

The nuclear and organellar genomes of four *Zygnema* strains (*Zygnema circumcarinatum* SAG 698-1b, UTEX 1559 and UTEX 1560 and *Z.* cf. *cylindricum* SAG 698-1a_XF; Fig. 1a–c) were assembled (Supplementary Table 1a–d). *Zygnema* cells are arranged in multicellular filaments containing two chloroplasts per cell (Fig. 1a,c,d) and much thinner (~400 nm) cross walls than outer walls (~1 μm; Fig. 1d,e and Supplementary Fig. 1), reflecting on their true filamentous body plan. Using chromatin conformation data (Dovetail Hi-C), we scaffolded the *Z. circumcarinatum* SAG 698-1b genome (N50, 4 Mb; Table 1 and Supplementary Table 1c) into 20 pseudo-chromosomes (Fig. 1f), which were supported by cytological chromosome counting[13] (Fig. 1b and Supplementary Fig. 1). The total assembly size (71 Mb) was close to sizes estimated by flow cytometry, fluorescence staining[14] and *k*-mer frequency analysis (Supplementary Fig. 2 and Supplementary Table 1b). The high mapping rates of UTEX 1559 and UTEX 1560 Illumina reads to the SAG 698-1b genome (97.16% and 97.12%, respectively) show that the overall genome structure was stable in the separate strain copies (Supplementary Text 1). UTEX 1559 and UTEX 1560 assemblies also have 20 pseudo-chromosomes. The three new *Z. circumcarinatum* genomes represent the first chromosome-level assemblies for any streptophyte alga (Table 1).

The nuclear genome assembly of SAG 698-1a_XF is five times larger (360 Mb) than those of *Z. circumcarinatum* (Table 1 and Fig. 1g). The marked genome size differences further support the notion that SAG 698-1a_XF and SAG 698-1b are two different species (Table 1 and Fig. 1a). Following a recent study[14], we refer to SAG 698-1a_XF as *Z.* cf. *cylindricum* (Fig. 1g, Table 1, Supplementary Table 1e,f and Supplementary Figs. 3–5).

### The smallest sequenced streptophyte genome

The three *Z. circumcarinatum* genomes reported here are the smallest among all streptophyte algae sequenced thus far (Table 1, Fig. 1g and Supplementary Table 1i). The genome of SAG 698-1b contains 23.4% repeats, while *Z.* cf. *cylindricum* SAG 698-1a_XF contains 73.3% (Supplementary Table 1j). No evidence for whole genome duplication (WGD) was found in *Zygnema* (Supplementary Fig. 6); *Z.* cf. *cylindricum* is probably polyploid (Supplementary Fig. 2).

Our phylogenetic analyses show that SAG 698-1b and UTEX 1560 are closer to each other than to UTEX 1559 (Fig. 1g and Supplementary Fig. 7). Gauch[15] reported that UTEX 1559 was a nonfunctional mating type (+) whereas UTEX 1560 and SAG 698-1b were functional mating type (−); indeed, our conjugation experiments failed to conjugate UTEX 1559 with UTEX 1560 or SAG 698-1b. Whole genome alignments (Supplementary Fig. 8) found chromosomes 20, 13 and 16 to differ the most among the three genomes, suggesting that they might contain sex/mating determination loci. *Zygnema* mating loci are so far unknown, and we did not identify homologs of the sex hormone proteins (protoplast release-inducing protein (PR-IP) and its inducer) described in *Closterium* (Supplementary Table 1k). The recently identified[8] *Cp*Minus1, an RWP-RK domain-containing protein that determines the mating type in heterothallic *Closterium*, does have homologs in *Zygnema*, but they are considerably longer (172 amino acids in *Cp*Minus1 versus 641 in Zci_02303 and 785 in Zci_08682). A total of 17,644 genes were shared by all three *Z. circumcarinatum* genomes (Supplementary Fig. 8f).

### Enriched orthogroups and domains in Zygnematophyceae

For the LCA of Zygnematophyceae + Embryophyta (Z + E) we infer an overrepresentation of Pfam domains (Fig. 2a,b), including (1) Chal_sti_synt_C (found in the key enzyme of the flavonoid pathway chalcone synthase), (2) Methyltransf_29 (found in *Arabidopsis* AT1G19430, required for cell adhesion[16]), (3) pentatricopeptide repeat (PPR) domains involved in organellar RNA binding and editing, and (4) domains related to plant immunity such as leucine-rich repeat (LRR) and Peptidase_S15, PK_Tyr_Ser-Thr and thioredoxins[17]; some overrepresentations could be gains via horizontal gene transfer (HGT), including Chal_sti_synt_C[18] and O-FucT (Supplementary Fig. 9). The Z + E LCA had enriched Gene Ontology (GO) terms related to biosynthesis of phytohormones, lipids and glucan (Fig. 2c), with 493 orthogroups (OGs) exclusive to Z + E (Fig. 2d), enriched in 'cation transmembrane transporter' and 'cell wall polysaccharide metabolic' (Fig. 2e).

A total of 3,409 Pfam domains were present in at least one representative of Cholorophyta, Embryophyta, Zygnematophyceae and

other streptophyte algae; 99 were exclusive to Z + E, and 27 to Zygnematophyceae (Fig. 2f). Some domains exclusive to Z + E could be the result of HGT (Supplementary Table 2). For example, Inhibitor_I9 and fn3_6 domains are among the most abundant in Z + E (Fig. 2g) and often co-exist with Peptidase_S8 domain in plant subtilases (SBTs; Fig. 2i), reportedly acquired from bacteria[19,20]; the same goes for the WI12 domain, named after the cell wall protein WI12 induced by diverse stressors[21] and key for pathogen defense[22] (Fig. 2g).

Combining existing protein domains is a powerful mechanism for functional innovation, as shown for cell adhesion, cell communication and differentiation[23] (Fig. 2h). A total of 982 Pfam domain combinations are shared by all studied genomes; 260 are unique to Z + E and 209 to Zygnematophyceae. Among those exclusive to Z + E (Fig. 2i), we found Lectin_legB and Pkinase domains (Supplementary Table 2) that were only combined into the same protein in the Z + E ancestor (despite individually having older evolutionary origins): for example, Zci_10218 (an L-type lectin receptor-like kinase (LecRLKs) family protein), featuring an extracellular Lectin_legB domain, an intracellular Pkinase domain and a middle transmembrane domain[24].

### Increased sophistication and resilience via expansions
We inferred 26 significantly expanded OGs in the LCA of Z + E (Fig. 2a and Supplementary Table 3a,b), three of which are related to phytohormone signaling[9,10,25]. Several expansions suggest more sophisticated gene networks featuring cornerstones in plant stress response and environmental signaling[26,27], and transmembrane transporters, including those involved in biotic interactions.

The LCA of Zygnematophyceae displayed 25 significantly expanded OGs (Fig. 2a and Supplementary Table 3b). Most expanded are alpha-fucosyltransferases (OG 89) involved in xyloglucan fucosylation[28]. We found genes encoding ethylene sensors and histidine kinase-containing proteins (OG 94), bolstering the idea that two-component signaling is important and active in filamentous Zygnematophyceae[29,30]. Several OGs were associated with typical terrestrial stressors; for example, Zygnema has an expected[31] set of phenylpropanoid enzyme-coding homologs (Supplementary Fig. 10). Several expanded OGs relate to development: expanded signaling and transport, possibly related to filamentous growth. A dynein-coding homolog (OG 72) was significantly contracted in Zygnematophyceae (Supplementary Table 3b), in line with the loss of motile gametes in Zygnematophyceae and OG 72 contraction in the Z + E ancestor.

Zygnema's stress resilience is renowned; it thrives in extreme habitats such as the Arctic[32]. We recover 16 significantly expanded OGs for the LCA of all four Zygnema strains (Fig. 2a and Supplementary Table 3b), including PP2C-coding genes (OG 548) often involved in abiotic stress signaling; the expansions of PP2Cs are shared among Zygnema spp. but independent of the radiation of PP2CAs in land plants (Supplementary Fig. 10). Along these lines, expanded OGs further included photoprotective early light-inducible proteins (ELIPs; OG 97)—probably the result of gene duplications in Zygnema cf. cylindricum (35 homologs versus 5 in Z. circumcarinatum SAG 698-1b or 2 in Arabidopsis thaliana)—and low-$CO_2$ inducible LciC's (OG 459). Like other

Zygnematophyceae[33,34], Zygnema has neochromes (Supplementary Fig. 11). Two OGs were significantly contracted: genes for GTP binding elongation factor Tu family (OG 251) and seven transmembrane MLO family protein (OG 320). On balance, the evolution of gene families reflects Zygnema's resilience in the face of challenging habitats.

The LCA of Z. circumcarinatum displays reduction of expanded OGs (Fig. 2a and Supplementary Table 3b) aligning with its genomic streamlining (see also Supplementary Figs. 12–16).

### Multicellularity and protein domain combinations
Our micrographs of Zygnema support previous descriptions[35], showing cells of a filament separated by very thin cross walls (Fig. 1d,e) that develop after cell division by cleavage, centripetally from the outside. Cells are surrounded by a homogalacturonan-rich extracellular matrix[36] (ECM; Fig. 1d), while Zygnema lack plasmodesmata, diverse cross cell walls have been described in Zygnematophyceae, including in filamentous Desmidiaceae[37]. Zygnema lacks rhizoids and rarely branches; short branching occurs in other Zygnematophyceae such as Zygogonium[38]. These observations indicate that true multicellularity occurs in Zygnematophyceae. Indeed, we infer for the LCA of Zygnematophyceae several expanded OGs related to development (Supplementary Table 3b). Expanded signaling and transport, which may relate to filamentous growth, include genes for calcium signaling (OG 56), zinc-induced facilitators (OG 258), cysteine-rich fibroblast growth factor receptors found in the Golgi apparatus (OG 518), and cation/$H^+$ antiporters (OG 809) related to AtNHX5/6 acting in pH and ion homeostasis in the endosome, key for membrane trafficking in the trans-Golgi network[39,40], and development by influencing auxin gradients[41].

There have been multiple gains and losses of multicellularity in Zygnematophyceae[6], but overall, it seems that gene gains are not the main drivers for multicellularity in filamentous Zygnema (Fig. 3a,b, Supplementary Table 4 and Supplementary Text 2). Significant domain expansions in multicellular streptophyte algae inlcude CHROMO (PF00385; particularly in Chara braunii), a domain integrating chromatin association with increased regulatory complexity[42] (Fig. 3c), F-box (PF00646), F-box like (PF12937), Myb_DNA-bind_4 (PF13837), Myb_DNA-bind_6 (PF13921), COesterase (PF00135) and LRR_4 (PF12799; Fig. 3c). Expansions in protein-coding genes for F-box and MYB TFs suggest diversified regulatory and signaling processes, including phytohormone signaling processes[30,43–47]. Despite these expansions, Pfam domain repertoires of unicellular and multicellular streptophyte algae showed 94% similarity (Fig. 3d).

Next, we investigated streptophyte protein domain combinations exclusive to multicellular algae and land plants compared with unicellular algae. The combination of EDR1, LRR_8 and Pkinase domains (PF14381, PF13855 and PF00069) probably evolved in the streptophyte LCA (Fig. 3e), occurring in the Arabidopsis putative Raf-related kinase (AT1G04210), involved in SnRK2 activation and osmotic stress response[48], E3 ubiquitin ligase interaction in regulating programmed cell death[49], and MAPK cascade activation[50]. The ubiquitin–homologous to the E6-AP carboxyl terminus (HECT) combination (PF00240 and PF00632) (Fig. 3f) occurs in Arabidopsis UPL5 (AT4G12570), which

**Fig. 2 | Comparative genomics of algal and land plant genomes. a**, Gene family expansion and contraction patterns estimated by CAFE using Orthofinder-identified OGs and the time-calibrated phylogeny of Fig. 1g. Key nodes are indicated on the tree and significant expansions and contractions are shown. The circles are proportional to expanded/contracted OGs; the numbers next to the circles indicate the numbers of expanded (orange) and contracted (dark gray) OGs. Z. cir., Zygnema circumcarinatum; Z. cyl., Zygnema cf. cylindricum. Icons indicate body plans: parenchymatous (box of tissue), filamentous (chain of cells), unicellular (single round cell) and sarcinoid/colonial (two round cells). **b**, Pfam domain enrichment for genes on the node leading to Zygnematophyceae and Embryophyta (Z + E). **c**, Functional (GO) enrichment for the Z + E node. **d**, OGs overlap among Chlorophyta, Embryophyta, Zygnematophyceae and other streptophyte algae. **e**, Enriched GO terms in the 493 OGs exclusive to Zygnematophyceae and Embryophyta. **f**, Pfam domain overlap among Chlorophyta, Embryophyta, Zygnematophyceae and other streptophyte algae. **g**, Exclusive Pfam domains found only in Zygnematophyceae and Embryophyta. One Pfam family WI12 was studied with phylogenetic analysis, suggesting a possible HGT from bacteria and expression response to stresses. **h**, Pfam domain combination overlap among Chlorophyta, Embryophyta, Zygnematophyceae and other streptophyte algae. **i**, Exclusive Pfam domain combinations in Zygnematophyceae and Embryophyta. Smu, Spirogloea muscicola; Pma, Penium margaritaceum; Men, Mesotaenium endlicherianum; SAG 698-1a_XF, SAG 698-1b, UTEX 1559 and UTEX 1560, the four here sequenced Zygnema spp.; Mpo, Marchantia polymorpha; Ppa, Physcomitrium patens; Ath, Arabidopsis thaliana.

is intertwined with intracellular signaling—featuring jasmonate and $H_2O_2$—in development and leaf senescence[51]. All multicellular algae and embryophyte species possess this domain combination in at least one ortholog. It thus probably dates back to a deep LCA, suggesting

secondary loss in unicellular algae. ARP8 (AT5G56180; Fig. 3g) stands out by combining F-box like (PF12937) and actin (PF00022) domains. It is involved in the ubiquitin E3 SCF complex, cell cycle regulation and chromatin remodeling via ubiquitin–proteosomal degradation[52]. This

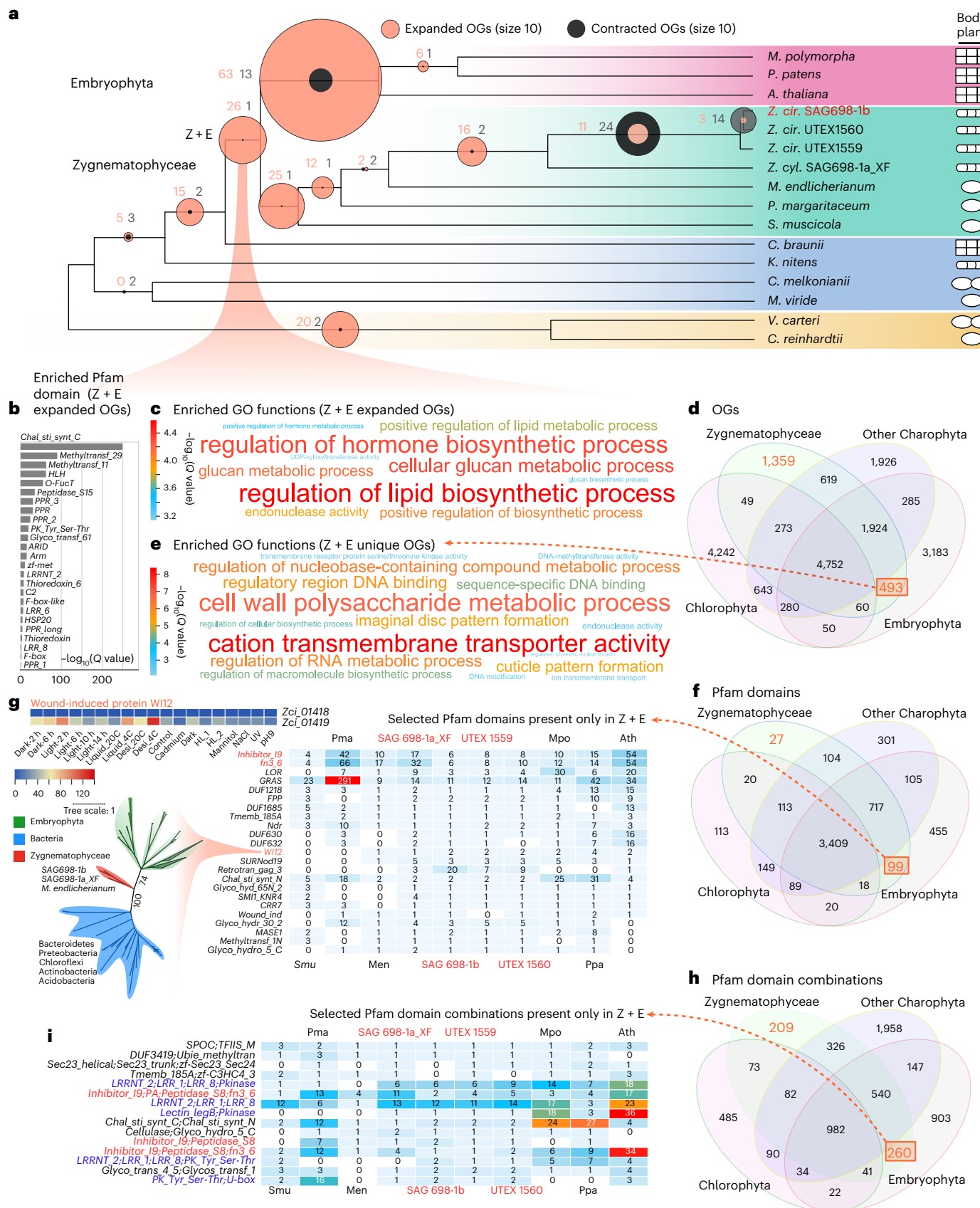

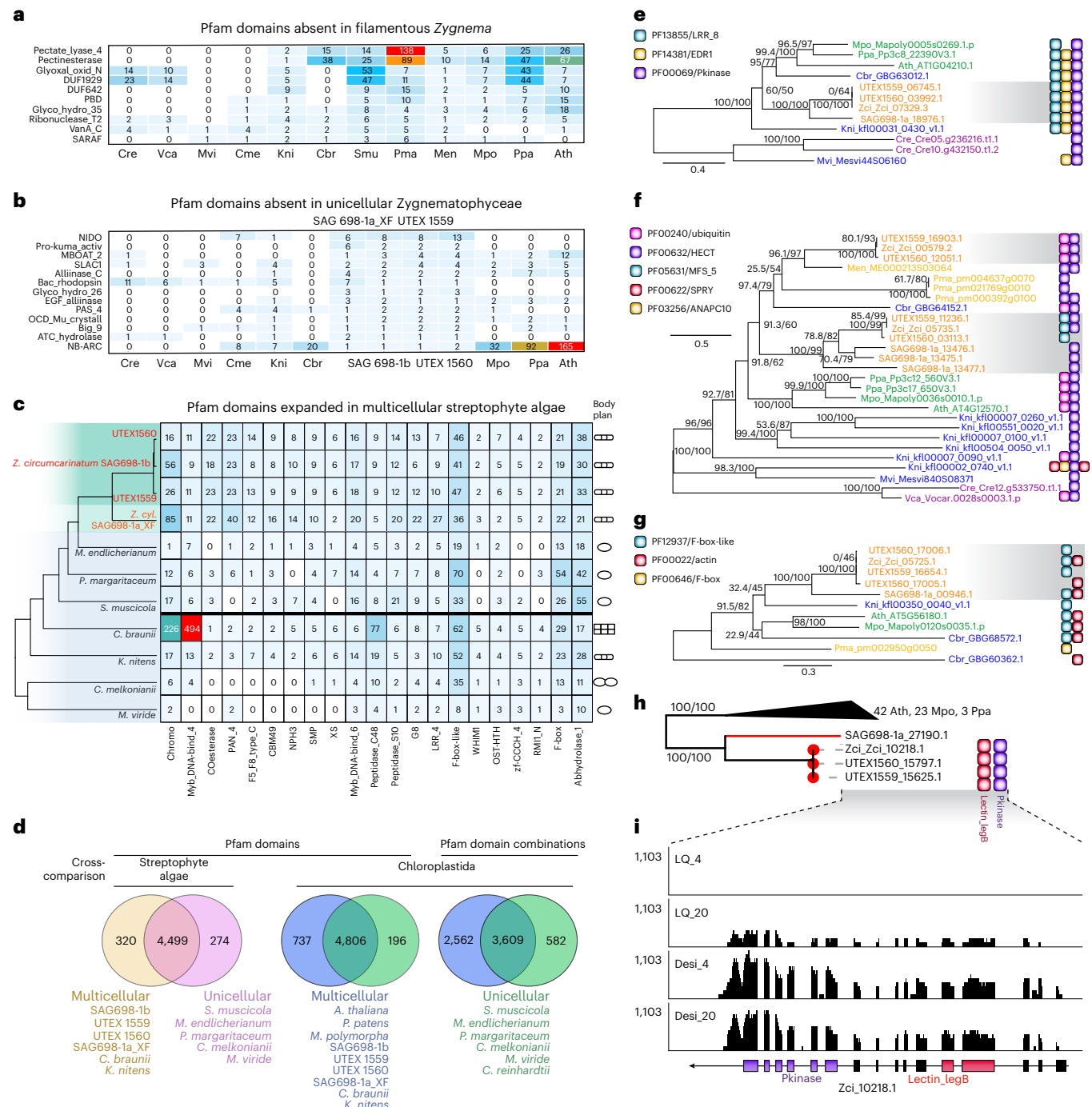

**Fig. 3 | Protein domains in unicellular and multicellular species in the green lineage. a**, Selected Pfam domains that are absent in the four filamentous *Zygnema* genomes. **b**, Selected Pfam domains that are absent in the three unicellular Zygnematophyceae genomes. **c**, A heatmap of selected Pfam domains that are significantly expanded in multicellular streptophyte algae; icons indicate body plans. **d**, Venn diagrams showing shared and exclusive Pfam domains and domain combinations in multicellular versus unicellular species. **e**–**g**, Phylogenetic trees of selected OGs and the corresponding protein domain architecture for each sequence. Phylogeny of Raf-related kinases bearing a combination of EDR1, LRR_8 and Pkinase domains (**e**). Phylogeny of HECT domain-containing ubiquitin protein ligases (**f**). Phylogeny of F-box-like

domain-containing actin-related proteins (**g**). **h**, Phylogeny of Zci_10218.1, a gene encoding L-type LecRLK with Lectin_legB domain in the N-terminus, Pkinase in the C-terminus and a TM domain in the middle. **i**, RNA-seq read mapping of Zci_10218.1. LQ, liquid; Desi, dessication; 4 and 20, temperature in Celsius; 1,103, the highest read counts (*y* axis). Cre, *Chlamydomonas reinhardtii*; Vca, *Volvox carteri*; Mvi, *Mesostigma viride*; Cme, *Chlorokybus melkonianii*; Kni, *Klebsormidium nitens*; Cbr, *Chara braunii*; Smu, *Spirogloea muscicola*; Pma, *Penium margaritaceum*; Men, *Mesotaenium endlicherianum*; SAG 698-1a, SAG 698-1b, UTEX 1559 and UTEX 1560, the four here sequenced *Zygnema* spp.; Mpo, *Marchantia polymorpha*; Ppa, *Physcomitrium patens*; Ath, *Arabidopsis thaliana*.

combination is exclusive to multicellular streptophyte algae and land plants and probably emerged in their LCA (Fig. 3g). A prominent combination particular for *Zygnema* is the Lectin_legB domain, one of the many

lectin families[53] important for plant immunity and development. We found Lectin_legB with other domains in 26 different combination architectures, often lineage-specific and differentially expressed (Fig. 3h,i).

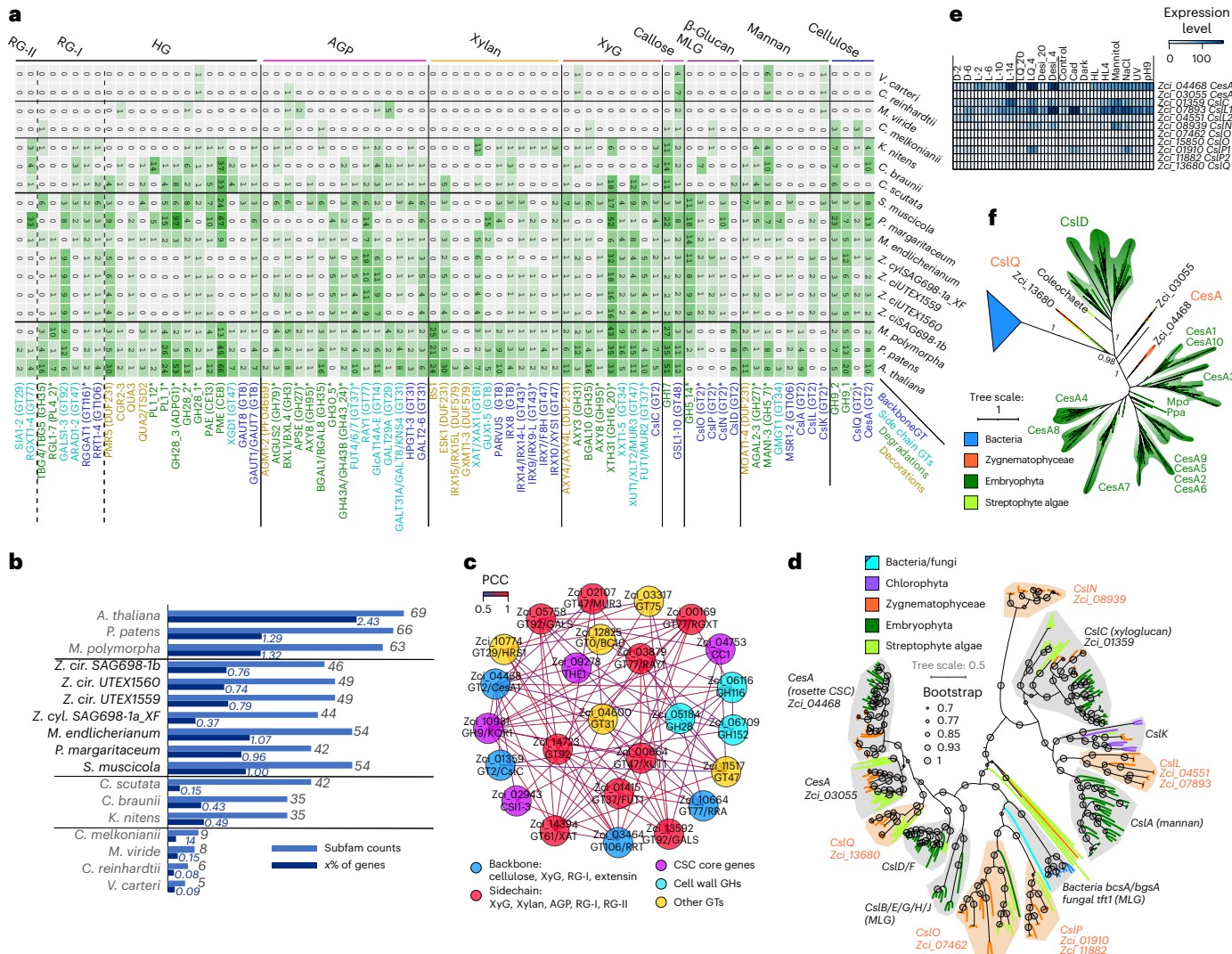

**Fig. 4 | Cell wall innovations revealed by protein family analyses. a**, A heatmap of homolog presence in 77 enzyme subfamilies (rows) across 17 plant and algal genomes (except for *Coleochaete scutata,* for which we used its transcriptome). The enzyme subfamilies are grouped by polysaccharide, and the colors indicate their biochemical roles; phylogenetic patterns compatible with gene gains that might have involved HGT are indicated with asterisks. **b**, Counts of subfamilies and gene percentages (with respect to the total annotated genes) across the 17 species. Shown in the plot is the gene percentage × 100. **c**, The co-expression network of SAG 698-1b containing 25 genes (most belonging to the 77 analyzed subfamilies) involved in cell wall polysaccharide syntheses. **d**, The phylogeny of GT2 across the 17 species. Major plant CesA/Csl subfamilies are labeled by the SAG 698-1b homolog, and newly defined subfamilies are in red. Ten bacterial β-glucan synthase (BgsA) and fungal mixed-linkage glucan (MLG) synthase (Tft1) homologs are included to show their relationships with plant CesA/Csl subfamilies. **e**, Gene expression of 11 SAG 698-1b GT2 genes across 19 experimental conditions (3 replicates each). **f**, The phylogeny of GT2 with ZcCesA1 (Zci_04468) homologs retrieved by BLASTP against the protein non-redundant (nr) database of the National Center for Biotechnology Information (NCBI) (E-value <1 × 10⁻¹⁰); colors follow **d**, and >5,000 bacterial homologs from >8 phyla are collapsed (blue triangle). See Supplementary Data 1–13 for details. *Z. cir./Z. ci, Zygnema circumcarinatum*; *Z. cyl., Zygnema* cf. *cylindricum*.

Key to the elaborate multicellular development of land plants are Type II MADS-domain (or MIKC-type) transcription factors (TFs), featuring a keratin-like (K) domain for forming floral quartet-like complexes (FQCs)[54,55]. The increase and diversification of these TFs is tightly associated with evolutionary novelties[55]. Each *Zygnema* genome encodes one MADS-domain TF. They form a clade (Supplementary Fig. 16) and lack K domains. In transcriptomes[4,56], however, we found MADS-box genes encoding a K domain in other Zygnematophyceae including also a *Zygnema* species, forming a second separate clade, that was apparently lost in the *Zygnema* species sequenced here (Supplementary Fig. 16). This suggests the presence of two MADS-domain TFs in the Zygnematophyceae ancestor: (1) an ancestral Type II without a K domain (probably unable to form FQCs), and (2) the MIKC type, with (in vitro) demonstrated ability to form FQCs[57].

Overall, several protein domain combinations that are exclusive to multicellular species seem associated with fine-tuned regulation of cell division and differentiation. New protein domain combinations might have arisen through gene fusions, many of which occurred already in the LCA of the green lineage (Chloroplastida). On balance, the number of specific genes and domain combinations was humble. These patterns align with proposed concepts on the evolution of multicellularity in green algae[58,59]. It is rather the regulation of a conserved set of genes that underpins multicellularity than a burst of novelty, combined with secondary losses. To such regulation, we turn later in this study.

## Gene gains facilitated major cell wall innovations

The cellulosic fibrils of the cell wall are a biophysical denominator in multicellular morphogenesis of plants, forming the first layer of

protection from the environmental stressors that also the earliest land plants had to overcome[60]. We reconstructed the evolutionary history of 38 cell wall-related enzyme families (Supplementary Table 1l), which were further split into 77 well-supported subfamilies (Fig. 4a and Supplementary Table 1m). Most subfamilies belong to carbohydrate active enzyme (CAZyme) families known for the synthesis and modifications of celluloses, xyloglucans, mixed-linkage glucans, mannans, xylans, arabinogalactan proteins (AGPs) and pectins (Fig. 4a and Supplementary Table 1l,m). Analyzing the 77 enzyme subfamilies (Supplementary Text 3 and Supplementary Data 1) revealed the following: (1) Z + E share all the major enzymes for the synthesis and modifications of the diverse polysaccharide components, including those for sidechains and modifications (Fig. 3b; 42–54 subfamilies in Zygnematophyceae versus 63–69 in Embryophyta). (2) Many of the enzymes for cell wall innovations, especially for polysaccharide backbone synthesis, have older evolutionary origins in the LCA of Klebsormidiophyceae and Phragmoplastophyta (Fig. 4b; 35–69 subfamilies versus 8–9 in Chlorokybophyceae and Mesostigmatophyceae). Many of such subfamilies are expanded in Zygnematophyceae (Fig. 4b; for example, GH16_20, GT77, CE8 and CE13 in Fig. 4a). (3) Genes involved in the syntheses of different cell wall polymers (backbones and sidechains) are co-expressed in SAG 698-1b (Fig. 4c). (4) Phylogenetic patterns suggest that some of the enzymatic toolbox for cell wall polysaccharide metabolism originated via HGT (Fig. 4a), pronounced for degradation enzymes (for example, GH5_7, GH16_20, GH43_24, GH95, GH27, GH30_5, GH79, GH28, PL1 and PL4), but it is also observed for GT enzymes. (5) Frequent gene loss creates scattered distributions of homologs in Streptophyta (Fig. 4a; for example, *Zygnema* lacks entire families or some subfamilies of GH5_7, GH35, GT29, GT8, CE8, GH28 and PL1).

We scrutinized the GT2 family, which contains major cell wall synthesis enzymes such as cellulose synthase (CesA) and Csl (CesA-like) for hemicellulose backbones (Fig. 4d and Supplementary Text 3). Among the 11 SAG 698-1b *CesA/Csl* homologs, Zc*CesA1* (Zci_04468), Zc*CslL1* (Zci_07893), Zc*CslC* (Zci_01359), Zc*CslN* (Zci_08939) and Zc*CslP1* (Zci_0910) are induced by various stresses[61] (Fig. 4e). The two *CesA* homologs in SAG 698-1b (Fig. 4d,f) and all other Zygnematophyceae homologs are (co-)orthologs of land plant *CesA*. Zc*CesA1* (Zci_04468) is co-expressed with four known plant primary cell wall cellulose synthase complex (CSC) component core genes: *KOR* (Zci_10931), *CC1* (Zci_04753), *CSI1* (Zci_02943) and *THE* (Zci_09278) (Fig. 4c). This extends previous observations[62] suggesting that co-expression of CSC component genes is evolutionarily conserved since the common ancestor of Zygnematophyceae and land plants.

Overall, the phylogenetic analyses of key cell wall enzymes (Supplementary Table 1l and Supplementary Text 3) highlight the importance of ancient HGTs contributing to evolutionary innovations of cell walls, similarly to what has been proposed for other traits[9,18].

## Co-expression connects environment and multicellular growth

We computed co-expression networks and searched for homologs related to (1) cell division and development, (2) multicellularity, (3) stress response, (4) transporters, (5) phytohormones (see also

Supplementary Figs. 17–20), (6) calcium signaling and (7) plant–microbe interaction (Supplementary Table 3). A total of 150 out of 406 modules showed co-occurrence of at least two such functional categories, the most frequent co-occurrence being plant–microbe interaction and calcium signaling, followed by plant–microbe interaction and stress (Fig. 5a). To understand the cohorts of genes that can establish the flow of information from external stimuli to the adjustment of internal programs, we additionally explored the above 150 modules for the layered system of (1) sensors, (2) signal transducers and (3) internal programs such as cell division and growth. Sensors co-express with transducers such as protein kinases and TFs (for example, modules 2, 20, 21, 23, 126, 147 and 173). Several such modules (Fig. 5b and Supplementary Text 4) contain *ELIP*s, coding for proteins that respond to light stimulus and can reduce photooxidative damage by scavenging free chlorophyll[63] under cold stress (module 21; Fig. 5c and Supplementary Fig. 21)—as shown for other streptophyte algae[10,64]—but also under high light (HL), expressed alongside a chaperon-coding gene and *PsbS* (key for NPQ; module 20). Module 38 features an *OLEOSIN* homolog, bolstering their importance in zygnematophytes[65].

Signal transduction and processing featured genes for kinases (calcium-dependent and LRR receptor-like kinases), PP2C and TFs (for example, modules 13, 57, 96, 107, 121, 130, 148, 161 and 170) and their frequent co-expression with well-known downstream genes for cell division (for example, modules 10, 22, 52, 87, 117, 128 and 179) and stress response (for example, modules 38, 74, 88, 90, 123 and 151). For example, module 87 features genes for calcium-dependent, cyclin-dependent and receptor-like protein kinases and Ras-related signaling proteins (for example, RAB GTPases) that are involved in cell growth, CHK histidine kinase of the cytokinin signaling network and downstream genes for cell growth and division such as microtubule-associated proteins, dynamins or kinesins. Module 38 features genes for SCR TF and protein kinases. These co-express with phytohormone genes of the abscisic acid (ABA) pathway (*ABA4* and *LUT2*), an auxin-response factor (*ARF10*) ortholog, and a gibberellin 20 oxidase (*GA20OX2*) homolog—despite the lack of gibberelins in *Zygnema*; all in addition to cell growth and division-related genes such as kinesin, transglutaminases and many photosynthesis-related genes.

An example for the tight link of calcium signaling and biotic interaction (Fig. 5a) is the co-expression of genes for LRR proteins with the calcium sensor and kinase (CPK; Zci_12352) in module 128 and CDPKs in module 117 (Supplementary Fig. 21). The most connected node in module 117 is an *LRR* and it also features *PP2C*s. While calcium signaling has recently been proposed to link plant pattern- and effector-triggered immunity[66,67], it is also important in mutualistic interactions[68].

The frequent overlaps between sensors and transducers and between transducers and downstream targets suggest a hierarchy where environmental cues are received, transmitted and processed, allowing a complex downstream response that integrates a variety of extrinsic and intrinsic signals. This aligns with the idea that the biology of plant cells hinges on a molecular information-processing network[69]. Our co-expression analyses recover joint action of genes for first sensing the environment and then modulating growth and stress response

**Fig. 5 | Gene co-expression modules and phylogenetic distribution of land plant signature specialized metabolism and TFs. a**, Heatmap of per-module co-occurrence frequencies among genes associated with plant–microbe (p–m) interaction, calcium signaling, stress, transporters, cell division and diverse phytohormones (see abbreviations below); based on 150 out of 406 total gene co-expression modules showing co-occurrence of at least two functional categories. **b**, Modules 20, 21, 38 and 87 discussed in the main text; node (gene) sizes are proportional to number of neighbors and edge (co-expression) widths are proportional to Pearson's correlation coefficient whereas colors are those of interconnected genes; egde gradient colors highlight the two dominant gene categories as indicated in the key. Font colors indicate genes' likely roles in establishing a flow of information. The full gene co-expression results can be

accessed in our online portal (https://zygnema.sbs.ntu.edu.sg/). The gene names are based on homology and the proteins they likely encode. **c**, The phylogenetic distribution of genes coding for proteins involved in phytohormone biosynthesis, signaling and phenylpropanoid biosynthesis. **d**, The phylogenetic distribution of genes coding for TFs. CK, cytokinin; ETH, ethylene; AUX, auxin; SL, strigolactone; JA, jasmonic acid; GB, gibberellic acid; SA, salicylic acid; BR, brassinosteroids; PPP, phenylpropanoid; TR, transcriptional regulators; PT, putative transcription-associated proteins. Note that the high number of homologs found in *Penium margaritaceum* are probably due to the large genome of 3.6 Gb and >50,000 annotated proteins. *Z. cir., Zygnema circumcarinatum; Z. cyl., Zygnema* cf. *cylindricum*.

mechanisms in *Z. circumcarinatum*. We interpret some of these joint actions as signatures for a homologous genetic network that dates back (at least) to an ancestor of Zygnematophyceae and land plants.

The symbiotic association with fungi was one of the key innovations that allowed plants to colonize land[70]. All four genes involved in symbiotic functions were found in *Zygnema*: *DMI2/SYMRK* pro-ortholog

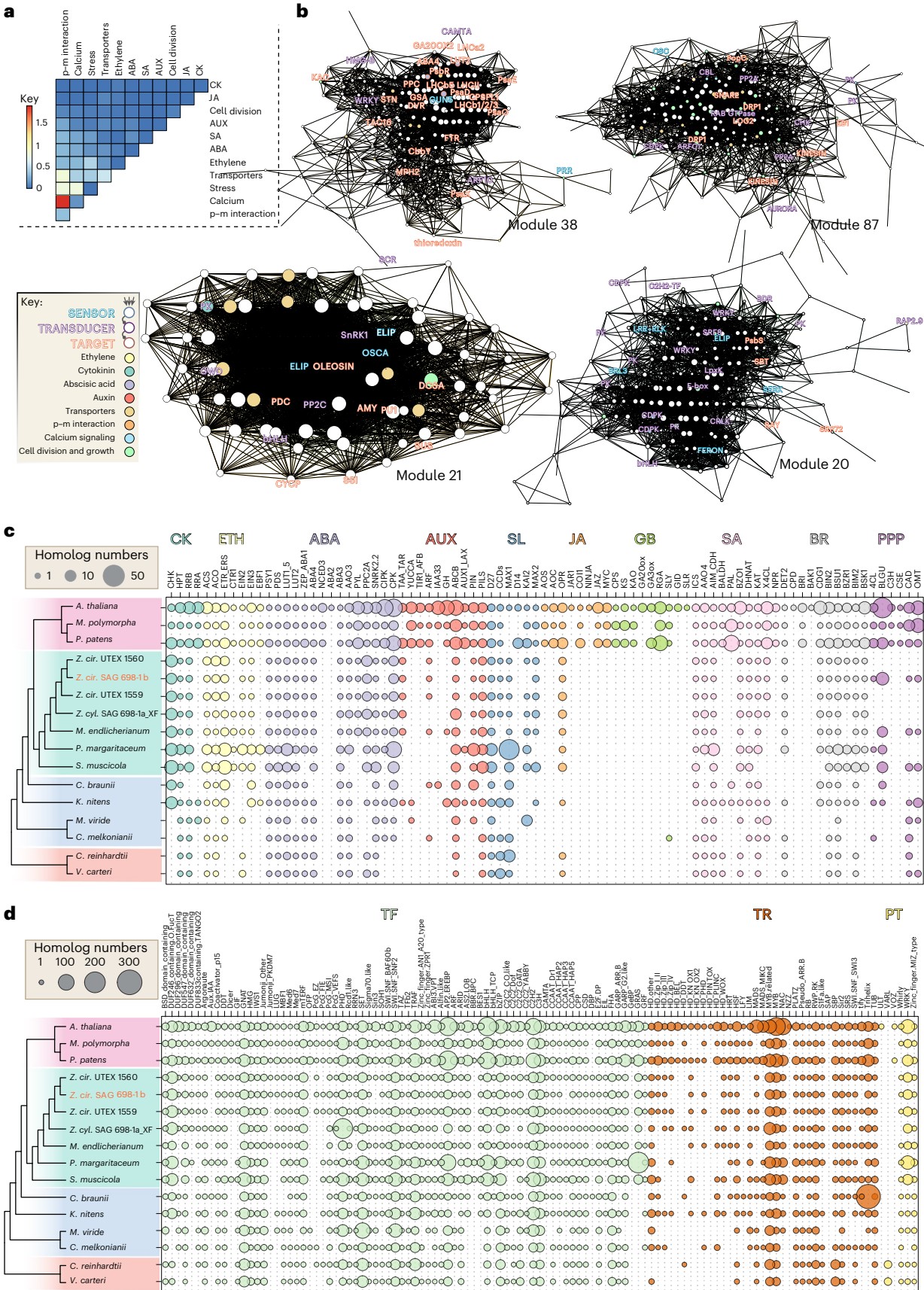

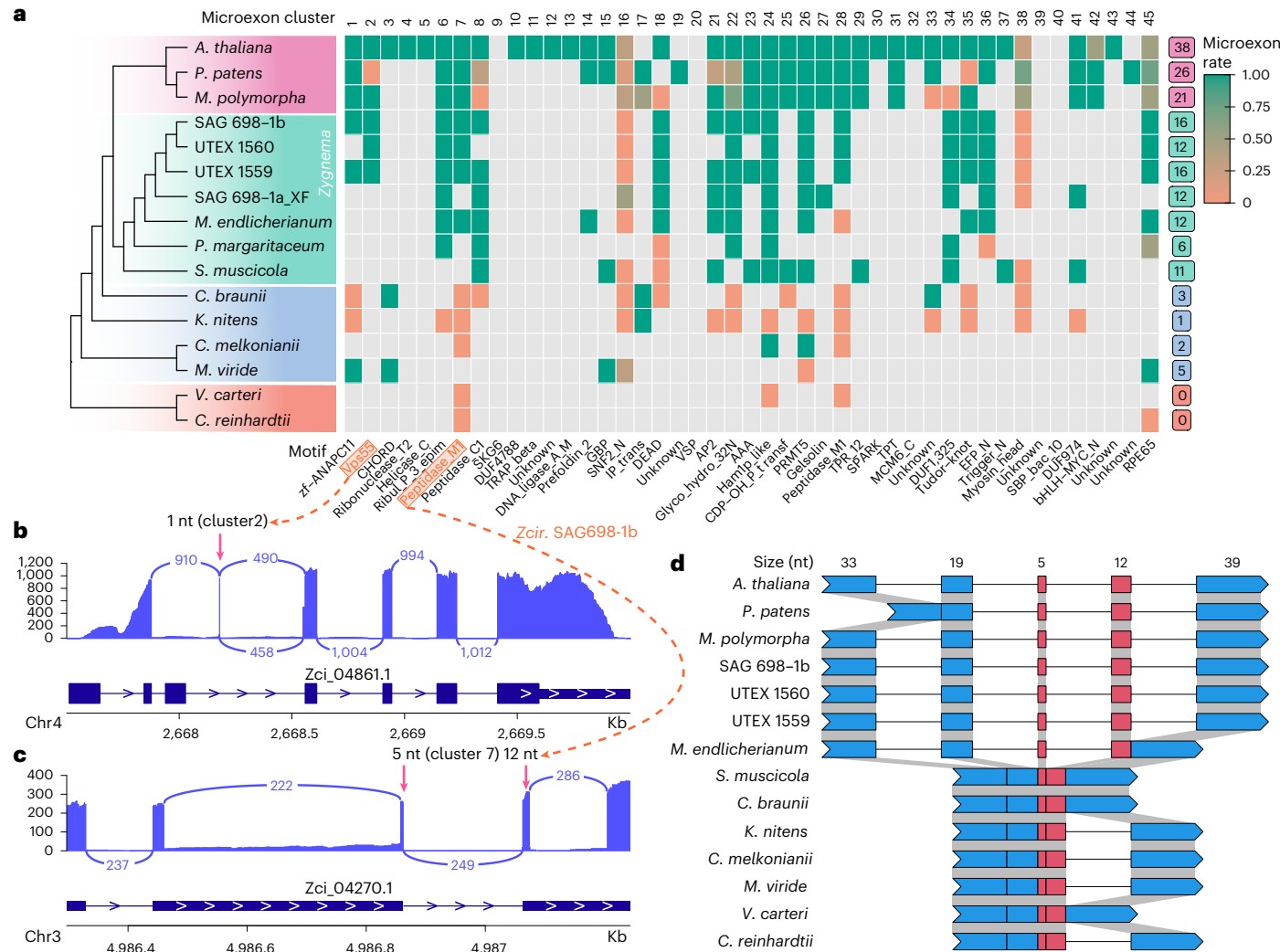

**Fig. 6 | Microexon prediction in 16 plant and algae genomes. a**, Heatmap of 45 conserved microexon-tags predicted by MEPmodeler. Microexon rate is the rate of true microexons among all predicted results in the cluster. For example, green cells indicate that 100% microexons with two flanking introns are present, orange indicates all microexon sequences are parts of large exons and none of them could be considered as microexons, and the others are between 0 and 1. A gray cell indicates missing data (a microexon-tag could not be found). The numbers on the right column indicate the predicted clusters containing at least one true microexon (see ref. 80 for more detail). **b**, RNA-seq evidence of the 1 bp microexon in cluster 2 (*x* axis the genomic location, and *y* axis the read count).

**c**, RNA-seq evidence of two adjacent microexons, 5 (cluster 7) and 12 bp (cluster 28). In **b** and **c**, the RNA-seq of condition p881sControl2 was used; RNA-seq read depth (blue numbers) and gene annotation are shown; blue arcs indicate introns (exon–exon junctions), and the numbers indicate the junction read counts supporting the introns. The pink arrows point to microexons. **d**, Exon–intron structures of microexon-tag clusters 7 and 28 in 14 plant genomes. The structure was predicted by relaxing the stringency in *M. viride* genome and by doing TBLASTX search in *S. muscicola* genome (all three copies are intronless in this microexon-tag), respectively. The others are predicted with default parameters.

(Zci_05951), *DMI1/POLLUX* (Zci_12099), *DMI3/CCaMK* (Zci_01672) and *IPD3/CYCLOPS* (Zci_13230; Supplementary Figs. 13–15). These genes belong in different modules (134, 78, 172 and 159, respectively), suggesting that the evolution of symbiosis in land plants recruited genes from diverse pathways rather than directly co-opting an existing pathway into a new function[71].

A comprehensive analysis of transcription-associated proteins (TAPs) with TAPscan[72] v.3 revealed higher numbers of TFs in land plants than in algae, as expected due to their more complex bodies and *Zygnema* species having comparatively more TAPs than other algae (Fig. 5d and Supplementary Table 3c). To further investigate the evolution of coordinated multicellular growth, we compiled a list of 270 genes with experimental evidence for roles in cell division (Supplementary Table 3d), finding that *Zygnema* lost microtubule plus tip proteins CLASP and SPIRAL1, potentially associated with the loss/reduction of rhizoids and phragmoplast-mediated cell division (cleavage instead).

Various gene modules (Supplementary Fig. 21) reflect cell division by co-expression of genes for proteins such as phragmoplastin (*DRP1*) (module 87), kinesin motors (for example, modules 52 and 87), spindle assembly (module 52; Supplementary Fig. 21), RAB GTPases (modules 10 and 87), SNARE (modules 52 and 87), cargo complex components (modules 10 and 87) and cell division-related kinases. Genes that probably originated in the Z + E LCA are *UGT1, SUN1/SUN2* and *LONESOME HIGHWAY*. The clearest cases of genes originating in the Z + E LCA code for GRAS TFs[9], including pro-orthologs of *SCARECROW* (*SCR*), *SCARECROW*-like and *SHORTROOT* (Supplementary Fig. 12), regulators of embryophyte cell division orientation and tissue formation—but also abiotic stress responses[73–76]. *Zygnema* GRAS homologs co-express with genes involved in cell division, cell cycle regulation and cell wall functions (modules 147, 38 (Fig. 5b) and 93). All three modules contain genes associated with abiotic stress responses, such as an *ELIP* homolog (OG 97; expanded in *Zygnema*), β-glucosidase (OG 85; expansion in

Z + E LCA), calcium cation channel (DMI1/POLLUX/CASTOR) and other calcium signaling components. The involvement of GRAS TFs in developmental and environmental signaling speaks of a complex network to coordinating growth and stress since the LCA of Z + E.

## Evolution of phytohormone pathways

Phytohormone biosynthesis and signaling networks have deep evolutionary roots. While gibberellins and jasmonates probably originated in land plants[30], other phytohormone pathways were at least partly present in algal ancestors (Fig. 5c and Supplementary Text 5). Land plants have more phytohormone-associated homologs than algae, as expected for their more complex signaling pathways[77], and Zygnematophyceae are overall similar to other streptophyte algae (Fig. 5c).

For example, despite the ABA biosynthesis pathway being incomplete, we detected $1.01 \pm 0.13$ ng g$^{-1}$ ABA in SAG 698-1b by liquid chromatography–mass spectrometry (Supplementary Fig. 17). The presence of diverse carotenoid cleavage dioxygenases (Supplementary Figs. 18–20) might point to alternative biosynthetic routes; perhaps via an ABA1-independent pathway starting upstream of zeaxanthin as suggested earlier[78]. Major aspects of the ABA signaling network are conserved across land plants[79]. The four new *Zygnema* genomes contain a complete set of homologous genes to the ABA signaling cascade, including the receptors, corroborating previous data on Zygnematophyceae[9,10]. Functional data showed that *Zc*PYL regulates PP2C in an ABA-independent manner[25].

## Microexons evolved during plant terrestrialization

Microexons (~1–15 bp) can be evolutionarily conserved and crucial for plant gene functions[80]. We predicted 45 microexon-tags in 16 plant genomes using MEPmodeler[80]. Land plants typically have >20 of 45 microexon-tag clusters. In Zygnematophyceae, we found 10–20 microexon-tag clusters (6 in *Penium margaritaceum* probably due to the fragmented genome assembly; Table 1), <5 in other streptophytes and none in Chlorophyta (Fig. 6). Zygnematophyceae and land plants have the most microexons. For example, a 1 bp microexon of cluster 2 was found in *Vps55* (Zci_4861) (Fig. 6b). Two adjacent microexons, 5 bp (cluster 7) and 12 bp (cluster 28) were found in a Peptidase M1 family gene (Zci_04270), which were overlooked by the de novo gene annotation (annotated as UTR and missed a Peptidase M1 motif) but verified by RNA sequencing (RNA-seq) (Fig. 6c). The two adjacent microexons are in the context of a 108 bp coding region spanning five exons in the *Arabidopsis* gene (AT1G63770.5). The five-exon structure is only conserved in land plants and *Zygnema* (Fig. 6d), whereas in *Mesotaenium endlicherianum* the last two exons (including the 112 bp) are fused, and all other algae have two or three exons with two adjacent microexons of clusters 7 and 28 always fused. It appears that, during terrestrialization, at least for this Peptidase M1 family gene, there was a gradual intronization process that created more microexons in land plants.

## Discussion

We generated chromosome-level genome assemblies for four filamentous algal sisters to land plants and performed comprehensive comparative genomics and co-expression network analyses. We found molecular innovations for signaling, environmental response and growth, and pinpoint their evolutionary history by tracing gene family expansions along the phylogenetic backbone of streptophytes. The reconstruction of ancestral gene content is a powerful means to explain the evolution of plant form and function as well as biological novelty[81]. Our data indicate the dynamics in Zygnematophyceae genome evolution (Fig. 2a), highlighting the need for a phylodiverse species set and the integration of complementary comparative approaches to understand the nature of the LCA of land plants and algae.

Throughout their evolutionary history, Zygnematophyceae have transitioned several times between multicellular and unicellular

body plans[6]. A parsimonious explanation is that streptophytes share an ancient toolkit for multicellularity[82,83], which comes to bear in filamentous genera but is still lingering as genetic potential in zygnematophyte unicells. And indeed, our data on shared OG expansions recover several important regulatory genes for increasing cellular complexity in the LCAs both of Z + E and of Zygnematophyceae. While we recover some specific protein domain gains, losses and combinations that might underpin actualization of filamentous growth, it appears more likely that the regulation of the shared toolkit for multicellularity is the critical factor in the evolution of filamentous algal bodies.

A defining feature of land plants is the plastic development of their multicellular bodies, ever adjusting to environmental conditions. High connectivity between genes involved in multicellularity and environmental stress response establishes the foundation for an adaptive advantage of multicellular morphogenesis, where cell differentiation can be fine-tuned for acclimation to environmental cues.

Genes that are co-expressed are often functionally related and concertedly act in genetic programs. We recover programs of an intrinsic nature, such as growth and development, cell division and cell wall biosynthesis/remodeling and genes that act in environmental sensing and signaling, triggered by an extrinsic input. In an interconnected module, there is an implicit directionality (outside/environment to inside). By their nature, signaling proteins must act in a genetic hierarchy (transduction through kinase cascades), and so do TFs (there must be an upstream and downstream). Both are co-expressed with intrinsic growth programs, thus revealing links between internal and external, suggesting joint actions of genes to sense the environment and modulate growth and reveal the genetic network underpinning molecular information processing in both plant and algal cells. This network has deep evolutionary roots, dating back at least to the ancestor of Z + E (Supplementary Text 4 and Supplementary Fig. 21).

Our data demonstrate a deep evolutionary origin of plant signaling cascades for acclimation to environmental cues and suggest a deep conservation of interconnections with regulation of growth—connections between extrinsic environmental input and intrinsic developmental programs that were drawn before Embryophyta began their conquest of land.

## Online content

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

[1]Nebraska Food for Health Center, Department of Food Science and Technology, University of Nebraska-Lincoln, Lincoln, NE, USA. [2]Institute of Microbiology and Genetics, Department of Applied Bioinformatics, University of Goettingen, Goettingen, Germany. [3]Campus Institute Data Science, University of Goettingen, Goettingen, Germany. [4]Section Phylogenomics, Centre for Molecular biodiversity Research, Leibniz Institute for the Analysis of Biodiversity Change, Zoological Museum Hamburg, Hamburg, Germany. [5]University of Nebraska-Lincoln, Center for Plant Science Innovation, Lincoln, NE, USA. [6]Nanyang Technological University, School of Biological Sciences, Singapore, Singapore. [7]Laboratoire de Recherche en Sciences Végétales, Université de Toulouse, CNRS, UPS, INP Toulouse, Castanet-Tolosan, France. [8]University of Jena, Matthias Schleiden Institute/Genetics, Jena, Germany. [9]Plant Cell Biology, Department of Biology, University of Marburg, Marburg, Germany. [10]Department of Algal Development and Evolution, Max Planck Institute for Biology Tübingen, Tübingen, Germany. [11]Institute for Mediterranean and Subtropical Horticulture 'La Mayora', Málaga, Spain. [12]Computational Biology, Department of Biology, Center for Biotechnology, Bielefeld University, Bielefeld, Germany. [13]Northern Illinois University, Molecular Core Lab, Department of Biological Sciences, DeKalb, IL, USA. [14]University of Innsbruck, Department of Botany, Research Group Plant Cell Biology, Innsbruck, Austria. [15]Department of Biochemistry and Molecular Biology, Dalhousie University, Halifax, Nova Scotia, Canada. [16]University of Goettingen, Albrecht-von-Haller-Institute for Plant Sciences, Experimental Phycology and Culture Collection of Algae at Goettingen University, Goettingen, Germany. [17]University of Applied Sciences Mittweida, Faculty of Applied Computer Sciences and Biosciences, Section Biotechnology and Chemistry, Molecular Biotechnology, Mittweida, Germany. [18]Universität Hamburg, Institute of Plant Science and Microbiology, Microalgae and Zygnematophyceae Collection Hamburg and Aquatic Ecophysiology and Phycology, Hamburg, Germany. [19]Genome Sequencing Center, HudsonAlpha Institute for Biotechnology, Huntsville, AL, USA. [20]Department of Energy Joint Genome Institute, Lawrence Berkeley National Laboratory, Berkeley, CA, USA. [21]Environmental Genomics and Systems Biology Division, Lawrence Berkeley National Laboratory, Berkeley, CA, USA. [22]Department of Plant and Microbial Biology, University of California Berkeley, Berkeley, CA, USA. [23]North China University of Science and Technology, Tangshan, China. [24]Boyce Thompson Institute, Ithaca, NY, USA. [25]Plant Biology Section, Cornell University, Ithaca, NY, USA. [26]University of Freiburg, Centre for Biological Signalling Studies (BIOSS), Freiburg, Germany. [27]The Hebrew University of Jerusalem, The Robert H. Smith Institute of Plant Sciences and Genetics in Agriculture, Rehovot, Israel. [28]University of Nebraska-Lincoln, School of Biological Sciences, Lincoln, NE, USA. [29]Department of Biology, East Carolina University, Greenville, NC, USA. [30]State Key Laboratory of Crop Stress Adaptation and Improvement, School of Life Sciences, Henan University, Kaifeng, China. [31]University of Goettingen, Goettingen Center for Molecular Biosciences, Goettingen, Germany. [32]Present address: Zhejiang Lab, Hangzhou, China. [33]Present address: Germplasm Bank of Wild Species, Kunming Institute of Botany, Chinese Academy of Science, Yunnan, China. [34]These authors contributed equally: Xuehuan Feng, Jinfang Zheng, Iker Irisarri, Jan de Vries, Yanbin Yin. ✉e-mail: devries.jan@uni-goettingen.de; yyin@unl.edu

## Methods

### Algal strains

*Z. circumcarinatum* SAG 698-1b and *Z.* cf. *cylindricum* SAG 698-1a were obtained from the Culture Collection of Algae at Göttingen University (SAG) (https://sagdb.uni-goettingen.de); from 698-1a, a single filament was isolated and used to establish a new culture that we coined 698-1a_XF and deposited at SAG. *Z. circumcarinatum* UTEX 1559 and UTEX 1560 were obtained from the UTEX Culture Collection of Algae at the University of Texas Austin (https://utex.org/). For the history of these strains, see Supplementary Text 1.

### Transmission electron microscopy

Transmission electron microscopy was essentially performed as previously described[84] using two independent cell cultures and each time ≥15 algal filaments. One-month-old cultures of *Zygnema circumcarinatum* (SAG 698-1b) and 3-month-old cultures of *Z.* cf. *cylindricum* (SAG 698-1a) were fixed in 2.5% glutaraldehyde (in 20 mM cacodylate buffer, pH 6.8) for 1.5 h and rinsed with 20 mM cacodylate buffer, embedded in 3% agarose and post fixed in 1% OsO$_4$ (in 20 mM cacodylate buffer) at 4 °C overnight and dehydrated in increasing ethanol concentrations. Samples were transferred in propylene oxide and embedded in modified Spurr's resin and sectioned with a Reichert Ultracut (Leica Microsystems). The ultrathin sections were stained with 2% uranyl acetate and Reynold's lead citrate. Transmission electron micrographs were taken on a Zeiss Libra 120 transmission electron microscope (Carl Zeiss AG) at 80 kV, which was equipped with a TRS 2k SSCCD camera and operated by ImageSP software (Albert Tröndle Restlichtverstärker Systeme).

### DNA and RNA sequencing

Detailed protocols for DNA and RNA extraction have been published elsewhere[14,61,85] and are, together with more details on genome and transcriptome sequencing and assembly, detailed in Supplementary Materials and Methods. For RNA-seq, we subjected *Z. circumcarinatum* SAG 698-1b to 19 growth and stress conditions, after which RNA-seq was obtained for the construction of a gene co-expression network. Stress and RNA-seq experiments were done in three baches. The first batch followed Pichrtová et al.[86] and de Vries et al.[10] with modifications. Three-week algae were subcultured in 12 flasks of liquid Bold's Basal Medium (BBM) with 0.02% L-arginine and grown for 2 weeks under standard conditions: 16 h/8 h of light/dark cycle at 20 °C and ~50 µmol photons m$^{-2}$ s$^{-1}$. Then, the algae were treated for 24 h under four conditions: (1) 20 °C in liquid medium (standard control), (2) 4 °C in liquid medium, (3) desiccation at 20 °C and (4) desiccation at 4 °C. Four treatments each with three replicates were performed. For desiccation treatments, algae were harvested using a vacuum filtration with Glass Microfiber Filter paper (GE Healthcare, 47 mm) and 20 µl of modified BBM (MBBM) was added on the filter paper. Papers with algae were then transferred onto a glass desiccator containing saturated KCl solution[86], and the desiccator was sealed with petroleum jelly and placed in the growth chamber under standard culture conditions. Cultures grown in liquid conditions were harvested using a vacuum filtration with Whatman #2 paper (GE Healthcare, 47 mm). After 24 h of treatment, the 12 samples were transferred into 1.5 ml Eppendorf tubes and immediately frozen in liquid nitrogen and stored in −80 °C. For the second batch (six diurnal experiments), the algae were grown with the same control conditions as the above mentioned (16 h/8 h of light/dark cycle, 20 °C, ~50 µmol of quanta per squared meter per second) and samples were collected every 4 h: (5) diurnal dark 2 h, (6) diurnal dark 6 h, (7) diurnal light 2 h, (8) diurnal light 6 h, (9) diurnal light 10 h and (10) diurnal light 14 h. For the third batch (nine stress experiments): SAG 698-1b was precultivated at 20 °C, 16 h/8 h light/dark cycle at 90 µmol photons per squared meter per second on a cellophane disks (folia Bringmann) for 8 days. For certain treatments (NaCl, mannitol and CadmiumCl) the culture was transferred to a new Petri dish where the medium was supplemented with the substances. Algae where then subjected to (11)

150 µM NaCl (Roth) for 24 h, (12) 300 mM mannitol (Roth) for 24 h, (13) 250 µM CadmiumCl (Riedel-de Haën AG) for 24 h, (14) dark treatment for 24 h, (15) high light (HL) treatment at 900 µmol photons per squared meter per second for 1 h, (16) ultraviolet-A at 385 nm, 1,400 µW cm$^{-2}$ for 1 h, (17) HL at 4 °C (HL4) at 900 µmol photons per squared meter per second for 1 h, (18) pH 9 for 24 h, and (19) a corresponding control growth at 20 °C on a plate.

### Library preparation and sequencing

The four genomes were sequenced by a combination of PacBio High-Fidelity (HiFi) long reads, Oxford Nanopore long reads and Illumina short reads (Supplementary Table 1a,b). DNA samples were sequenced at the Roy J. Carver Biotechnology Center (University of Illinois, Urbana-Champaign) using Oxford Nanopore and Illumina technologies (Supplementary Table 1a). Oxford Nanopore DNA libraries were prepared with 1D library kit SQK-LSK109 and sequenced with Spot-ON R9.4.1 FLO-MIN106 flowcells for 48 h on a GridIONx5 sequencer. Base calling was performed with Guppy v1.5 (https://community.nanoporetech.com). Illumina shotgun genomic libraries were prepared with the Hyper Library construction kit (Kapa Biosystems, Roche). Libraries' fragment size averaged at 450 bp (250–500 bp) and were sequenced with 2×250 bp paired-end reads on a HiSeq 2500. Additional DNA samples were sequenced at the Genome Research Core (University of Illinois, Chicago) and Joint Genome Institute (JGI; Berkeley, California). The Illumina shotgun genomic libraries were prepared with the Nextera DNA Flex Library Prep Kit. Fragment sizes averaged at 403 bp and were sequenced with 2 × 150 bp paired-end reads on HiSeq 4000 (Supplementary Table 1a). RNA samples were sequenced at the Genome Research Core (University of Illinois, Chicago). The libraries were prepared by ribosomal RNA (rRNA) depletion with Illumina Stranded Total RNA kit plus Ribo-Zero Plant[87], and 2 × 150 bp paired-end sequencing was performed on HiSeq 4000. RNA from the third batch of stress experiments were sequenced at the NGS-Integrative Genomics Core Unit of the University Medical Center Göttingen, Germany. Stranded messenger RNA libraries were prepared with the Illumina stranded mRNA kit, and paired-end sequencing of 2×150 bp reads was carried out on an Illumina HiSeq 4000 platform. RNA-seq data for SAG 698-1a and UTEX 1559 have been previously published (Supplementary Table 1a).

### Genome assembly and scaffolding

To assemble the genome of SAG 698-1b, a total of 5.4 Gb (82×) of Oxford Nanopore nuclei DNA reads were assembled with wtdbg (v2)[88,89]. Assembled contigs were polished by Racon[90] and three iterations of pilon[91] with Illumina paired-end reads. The polished genome was scaffolded by Dovetail Genomics HiRise software with Hi-C sequencing data (https://dovetailgenomics.com/). Genome contamination was examined by BLASTX against NCBI's non-redundant (nr) database, and contaminated scaffolds were removed.

To assemble the UTEX 1559 genome, an initial assembly was done with SPAdes[92] using Illumina paired-end reads (2 × 150 bp), three mate-pair libraries (insert size 3–5 kb, 5–7 kb and 8–10 kb) and Oxford Nanopore reads (Supplementary Table 1a). Assembled contigs were further scaffolded by two rounds of Platanus-allee[93] with Illumina paired-end reads (2 × 250 bp), three mate-pair libraries (insert size 3–5 kb, 5–7 kb and 8–10 kb) and Oxford Nanopore reads. For the UTEX 1560 genome, Illumina paired-end (2 × 150 bp) and PacBio HiFi reads were used for assembly with SPAdes and further scaffolded with Platanus-allee. Scaffolds with contaminations were identified by BLASTX against NR and removed. The genomes of UTEX 1559 and UTEX 1560 were scaffolded by Dovetail Genomics HiRise software with Hi-C sequencing data from SAG 698-1b.

The genome of SAG 698-1a_XF was sequenced with PacBio HiFi long reads (40 Gb), Nanopore long reads (4 Gb) and Illumina short reads (>100 Gb). The *k*-mer analysis using Illumina reads revealed two peaks in the *k*-mer distribution, suggesting that SAG 698-1a_XF exists

as a diploid organism with an estimated heterozygosity rate of 2.22% (Supplementary Fig. 2). All Illumina short reads and the Nanopore reads were first assembled into contigs using SPAdes. Then, WENGAN[94] was used to assemble HiFi long reads and Illumina paired-end reads (2 × 150 bp) using the SPAdes contigs as the reference. Lastly, the resulting WENGAN contigs were scaffolded and gaps were closed with Platanus-allee using all the Nanopore, HiFi and Illumina reads to derive a consensus pseudo-haploid genome.

To evaluate the quality of assembled genomes (Supplementary Table 1d), raw RNA-seq reads, Oxford Nanopore and Illumina DNA reads were mapped to the assembly with hisat (v2)[95], minimap (v2)[96] and bowtie (v2)[97], respectively. To assess genome completeness, a BUSCO[98] analysis was performed with the 'Eukaryota odb10' and 'Viridiplantae odb10' reference sets.

### Genome annotation

In all four genomes, protein coding genes were predicted by the MAKER-P pipeline[99], which integrates multiple gene prediction resources, including ab initio prediction and homology- and transcripts-based evidence. First, repetitive elements were masked by RepeatMasker with a custom repeat library generated by RepeatModeler. Rfam with infernal and tRNA-Scan2 were used to analyze noncoding RNA and transfer RNA (tRNA). For the transcript evidence, a total of 103,967 transcripts were assembled by Trinity (reference-free) and StringTie (reference-based) from the respective RNA-seq data. Transcriptome assembly was used to generate complete protein-coding gene models using the tool Program to Assemble Spliced Alignments (PASA). Proteins from *Mesotaenium endlicherianum*, *Spirogloea muscicola* and *Arabidopsis thaliana* (TAIR10) were used for homology-based evidence. Then, the resulting protein-coding gene models from the first iteration of the MAKER-P pipeline were used as the training data set for SNAP and Augustus models, which were fed into MAKER for the second iteration of annotation. After three rounds of gene prediction, MAKER-P combined all the protein-coding genes as the final annotated gene set.

### Plastome and mitogenome assembly and annotation

NOVOPlasty 3.8.2 (refs. 100,101) was used to assemble plastomes. The contiguity of assembled plastomes was examined in Geneious (https://www.geneious.com/)[102] with read mapping. For SAG 698-1b mitogenome assembly, Oxford Nanopore reads were assembled with Canu[103], where one long mitogenome contig of 238,378 bp was assembled. This contig was circularized and polished with three rounds of pilon[91], which was further corrected with Illumina raw reads and compared with the mitogenome of UTEX 1559 (MT040698)[85] in Geneious. For SAG 698-1a_XF, PacBio HiFi reads were used for the assembly of its mitogenome.

Plastome and mitogenome annotation was performed with GeSeq[104,105]. For plastome annotation, BLAT search and HMMER profile search (Embryophyta chloroplast) were used for coding sequence, rRNA and tRNA prediction; ARAGORN v1.2.38, ARWEN v1.2.3 and tRNAscan-SE v2.0.5 were used for tRNA annotation. For mitogenome annotation, Viridiplantae was used for BLAT reference sequences. The annotated gff files were uploaded for drawing circular organelle genome maps on OGDRAW[106,107].

The plastome of SAG 698-1b is identical to those of UTEX 1559 (GenBank ID MT040697)[85] and UTEX 1560. The mitogenomes of SAG 698-1b (OQ319605; Supplementary Fig. 3) and UTEX 1560 are identical in sequence but slightly longer than that of UTEX 1559 (MT040698, 215,954 bp)[85] (Supplementary Fig. 4). The plastome of SAG 698-1a was available[108]. Its mitogenome (OQ316644) (Supplementary Fig. 5), at 323,370 bp size, is the largest known among streptophyte algae (Supplementary Table 1g,h).

### Repeat annotation and analysis

Repetitive DNA was annotated using the homology strategy with repeat libraries generated with RepeatModeler. RepeatModeler integrates RepeatScout, RECON, LTRharvest and LTR_retriever tools (version 2.0.1; refs. 109,110). The miniature inverted-repeat transposable elements (MITE) library was identified with MITE-tracker[111]. These two identified libraries were combined and incorporated into RepeatMasker (v.4.0.9; http://www.repeatmasker.org/) for repeat annotation. SAG 698-1b contains mostly simple repeats (6.4%) and transposable elements of the MITE (4.3%), Gypsy (2.9%) and Copia (1.9%) families. The *Z*. cf. *cylindricum* SAG 698-1a_XF genome has Copia (29.8%), MITE (11.6%), Gypsy (5.9%) and simple repeats (2.1%)

### Comparative genomics analysis

Sixteen representative genomes were selected, including chlorophytes (*Volvox carteri*[58] and *Chlamydomonas reinhardtii*[112]), Zygnematophyceae (*Z. circumcarinatum* SAG 698-1b, UTEX 1559, UTEX 1560, *Z*. cf. *cylindricum* SAG 698-1a_XF, *Mesotaenium endlicherianum*[9], *Penium margaritaceum*[7] and *Spirogloea muscicola*[9]), additional streptophyte algae (*Chara braunii*[11], *Klebsormidium nitens*[113], *Chlorokybus melkonianii*[114,115] (a strain formerly known as *C. atmophyticus*) and *Mesostigma viride*[114]), bryophytes (*Marchantia polymorpha*[116] and *Physcomitrium patens*[117]) and a vascular plant (*Arabidopsis thaliana*[118]).

OGs were inferred with Orthofinder[119]. Time-calibrated species phylogeny was built with low-copy OGs (≤3 gene copies per species). Divergence time estimation was carried out with MCMCTree. Expanded and contracted gene families were identified with CAFE and the species phylogeny. For microexon analyses, MEPmodeler[80] was used[120].

For comparative genomics analyses of multicellularity, the 16 genomes were classified into two groups, unicellulars (*Chlamydomonas reinhardtii*, *Chlorokybus melkonianii*, *Mesostigma viride*, *Spirogloea muscicola*, *Mesotaenium endlicherianum* and *Penium margaritaceum*) and multicellulars (*Volvox carteri*, *Klebsormidium nitens*, *Chara braunii*, SAG 698-1a_XF, SAG 698-1b, UTEX 1559 and UTEX 1560, *Marchantia polymorpha*, *Physcomitrium patens* and *Arabidopsis thaliana*). Proteins in the 16 genomes were annotated against the Pfam database to find functional domains. Domain occurrences (presense/absence) and abundances in each genome were recorded and compared between the two groups to infer domain gain, loss and combination.

Comparative genomics were performed with 16 representative green algal and plant genomes. Annotated proteins were clustered into OGs by OrthoFinder. A total of 4,752 OGs contained proteins from at least one representative of Chlorophyta, Embryophyta, Zygnematophyceae and other streptophyte algae (Fig. 2a–d).

A total of 1,359 OGs were Zygnematophyceae specific, with enriched GO terms 'phosphorylation', 'pyrophosphatase activity', 'transmembrane receptor protein serine/threonine kinase activity', 'cellular response to abscisic acid stimulus' and 'polysaccharide biosynthetic process', speaking of an elaboration of the molecular chassis for signaling cascades and cell wall biosynthesis.

We inferred expanded and contracted gene families with CAFE[121] using OGs from Orthofinder[119]. Among the 24 significantly contracted OGs are the light-harvesting complex (OG 43), ELIPs (OG 97; expanded in the *Zygnema* ancestor), RuBisCO small chain protein (OG 57), cell wall-related proteins such as expansins (OG 20), glycosyl transferases (OG 115) or glycoproteins (OG 182). The 11 expanded OGs feature lipases (OG 319), an uncharacterized protein with a methyltransferase domain (OG 637), a selenoprotein with a possible antioxidant activity (OG 1159), plant–microbe interaction proteins (OG 1170) and TFs (OG 777 and OG 1250).

We investigated protein domains and domain combinations that are gained, lost and significantly expanded in multicellular streptophyte algae. The top families present in filamentous *Zygnema* but absent in the three investigated unicellular Zygnematophyceae (Fig. 3b) include nidogen homology sequence (NIDO), which is present in animal glycoproteins but absent in land plants; Pro-kuma_activ, which corresponds to Peptidase S53, MBOAT_2, a domain in Wax synthase, involved in drought resistance; Alliinase, which is involved in auxin

biosynthesis; Bac_rhodopsin, which is present in light-dependent ion pumps and sensor proteins; Glyco_hydro_26, which is present in β-mannanase; and the NB-ARC domain known from plant disease resistance gene families. For most of these families, gene loss in unicellular Zygnematophyceae is more likely than a gain in filamentous *Zygnema*, because they are present in algae outside of Phragmoplastophyta. Exceptions are the peptidase Pro-kuma_activ and the β-mannanase domain Glyco_hydro_26.

### CAZyme and gene family phylogenetic analysis
CAZyme families were identified with dbCAN2 (ref. 122) with default parameters (E-value <1$^{-10}$ and coverage >0.35). Whenever needed, dbCAN2 was rerun by using more relaxed parameters. The experimentally characterized cell wall enzymes were manually curated from the literature (Supplementary Data 1 and Supplementary Table 1l). Reference genes were included into the phylogenies to infer the presence of orthologs across the 16 genomes and guide the split of large families into subfamilies. Phylogenetic trees were built by using FastTree initially, and for some selected families, RAxML[123] and IQ-TREE[124] were used to rebuild phylogenies to verify topologies.

### Co-expression network
The highest reciprocal rank co-expression network for *Z. circumcarinatum* (SAG 698-1b) was built from all RNA-seq samples (19 growth conditions), and the *Zygnema* database was established using the CoNekT framework[125]. The gene co-expression clusters were identified using the Heuristic Cluster Chiseling Algorithm with standard settings[126].

We explored functional gene modules in *Z. circumcarinatum* SAG 698-1b by inferring gene co-expression networks from RNA-seq data of 19 growth conditions (see above). We obtained 406 clusters (modules) containing 17,881 out of the 20,030 annotated gene isoforms. Candidate genes were drawn from the literature and the set of expanded OGs.

### Statistics
To identify possible WGDs, Ks and 4dtv values were calculated for each genome. First, all paralog pairs were identified using the Reciprocal Best BLAST Hit (RBBH) method using protein sequences (E-value <1 × 10$^{-6}$), following the method described by Bowman et al.[116]. RBBH paralog pairs were aligned with MAFFT[127], and the corresponding nucleotide alignments were generated. Using RBBH alignments of paralog pairs, KaKs_Calculator2.0 (ref. 128) with the YN00 model and the calculate_4DTV_correction.pl script were run to calculate Ks and 4dtv values for each alignment, respectively. Values with Ks of 0 and 4dtv of 0 were filtered. The Ks and 4dtv distributions were fitted with a Gaussian kernel density model using the seaborn package. For the SAG 698-1b chromosome-level genome, MCscan[129] was run to identify syntenic block regions with default parameters.

For the species phylogeny and divergence time analysis, a phylogenetic tree was built using RAxML v.8 (ref. 123) with the '-f a' setting and the PROTGAMMAJTT model, and branch support with 100 pseudoreplicates of nonparametric bootstrap. The tree was rooted with Chlorophyta as outgroup. Using the above methodology, additional phylogenetic analyses were performed with (1) the four *Zygnema* strains and (2) the seven Zygnematophyceae genomes, to obtain a higher number of single-copy loci, 5,042 and 204, respectively (Supplementary Fig. 7). Divergence time estimation was carried out with MCMCTree implemented in the PAML package version 4.10.0j (ref. 130). The 493 low-copy OG protein sequence alignment was converted to the corresponding nucleotide alignment for MCMCTree, in which ten Markov chain Monte Carlo (MCMC) chains were run, each for 1,000,000 generations (Supplementary Table 1f). Three calibration were set in the reference tree according to Morris et al.[131], on the nodes Viridiplantae (972.4 to 669.9 Ma), Streptophyta (890.9 to 629.1 Ma) and Embrophyta (514.8 to 473.5 Ma).

OG expansion and contraction were inferred with CAFE v.5 (ref. 121) using OGs inferred with Orthofinder[119] v.2.4.0 and the previously inferred time-calibrated species phylogeny. CAFE v.5 was run with default settings (base) using the inferred OGs and a calibrated species phylogeny. Two independent runs arrived to the same final likelihood and lambda values. The first eight OGs (OG0–7) were excluded from the analysis due to too drastic size changes between branches that hampered likelihood calculation; excluded OGs were mostly exclusive to a single *Zygnema* or *Chara* genome and probably represented transposable elements, as judged by results of BLASTP against NR.

The gene co-expression clusters were identified using the Heuristic Cluster Chiseling Algorithm with standard settings[126].

For phylogenies of gene families related to symbiosis, tree reconstruction was performed using IQ-TREE v2.1.2 (ref. 132) based on the Bayesian Information Criterion (BIC)-selected model determined by ModelFinder[133]; branch supports was estimated with 10,000 replicates each of both SH-aLRT[134] and UltraFast bootstraps[135]. For the other phylogenies, homologs were aligned with MAFFT v7.453 using the L-INS-I approach[127] and maximum likelihood phylogenies computed with IQ-TREE (v.1.5.5)[124], with 100 nonparametric bootstrap pseudoreplicates and BIC-selected model with ModelFinder[133].

### Reporting summary
Further information on research design is available in the Nature Portfolio Reporting Summary linked to this article.

### Data availability
The four *Zygnema* genomes, raw DNA reads and rRNA-depleted RNA-seq of SAG 698-1b can be accessed through NCBI BioProject PRJNA917633. The raw DNA read data of UTEX 1559 and UTEX 1560 sequenced by the Joint Genome Institute can be accessed through BioProjects PRJNA566554 and PRJNA519006, respectively. RNA-seq data of UTEX 1559 can be accessed through BioProject PRJNA524229. Poly-A enriched RNA-seq data of SAG 698-1b can be accessed through BioProject PRJNA890248 and the Sequence Read Archive (SRA) under the accession SRR21891679 to SRR21891705. *Zygnema* genomes are also available through the PhycoCosm portal[136] (https://phycocosm.jgi.doe.gov/SAG698-1a (ref. 137), https://phycocosm.jgi.doe.gov/SAG698-1b (ref. 138), https://phycocosm.jgi.doe.gov/UTEX1559 (ref. 139) and https://phycocosm.jgi.doe.gov/UTEX1560 (ref. 140)). Data files are available via Figshare at https://doi.org/10.6084/m9.figshare.22568197 (ref. 141) and via Mendeley at https://doi.org/10.17632/gk965cbjp9.1 (ref. 142). Source data are provided with this paper.

### Code availability
No original code was used; all computational analyses were performed with published tools and are cited in Methods and Supplementary Materials and Methods.

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

## Acknowledgements

This work was funded by the US National Science Foundation (NSF) CAREER award (DBI-1652164, 1933521), the Nebraska Tobacco Settlement Biomedical Research Enhancement Funds as part of a startup grant of the University of Nebraska Lincoln, the Research & Artistry Award of Northern Illinois University, the Joint Genome Institute Community Science Program (CSP), the United States Department of Agriculture (USDA) award (58-8042-9–089), the National Institutes of Health (NIH) awards (R21AI171952) and (R01GM140370) all to Y.Y., and by the German Research Foundation grant 440231723 (VR 132/4-1) to J.d.V., TH417/12-1 to G.T. and F.R., and 440540015 (BU 2301/6-1) to H.B. within the framework of the Priority Programme 'MAdLand – Molecular Adaptation to Land: Plant Evolution to Change' (SPP 2237), and grant 410739858 in the frame of the project CharMod to K.v.s., as well as RE 1697/16-1 (CharMod) and 18-1 (CharKeyS) to S.A.R. J.d.V. further thanks the European Research Council for funding under the European Union's Horizon 2020 research and innovation program (grant agreement no. 852725; ERC-StG 'TerreStriAL'). The work was further supported by Austrian Science Fund (FWF) project 10.55776/P34181 to A.H. The work (proposal: 10.46936/10.25585/60001088) conducted by the US Department of Energy Joint Genome Institute (https://ror.org/04xm1d337), a DOE Office of Science User Facility, is supported by the Office of Science of the US Department of Energy operated under contract no. DE-AC02-05CH11231. P.-M.D is supported by the project Engineering Nitrogen Symbiosis for Africa (ENSA) currently funded through a grant to the University of Cambridge by the Bill & Melinda Gates Foundation (OPP1172165) and the UK Foreign, Commonwealth and Development Office as Engineering Nitrogen Symbiosis for Africa (OPP1172165), by the 'Laboratoires d'Excellence (LABEX)' TULIP (ANR-10-LABX-41)' and by the European Research Council (ERC) under the European Union's Horizon 2020 research and innovation program (grant agreement no. 101001675). A.D., A.D.A., M.J.B. and J.M.S.Z. are grateful for being supported through the International Max Planck Research School (IMPRS) for Genome Science; J.M.R.F.-J. and T.P.R. gratefully acknowledge support by the Ph.D. program 'Microbiology and Biochemistry' within the framework of the 'Göttingen Graduate Center for Neurosciences, Biophysics, and Molecular Biosciences' (GGNB) at the University of Goettingen. The authors thank L. Pfeifer (University of Kiel) for excellent comments on the cell wall-related genes and E. Woelken (Universität Hamburg) for skillful technical assistance with transmission electron microscopy. The authors thank Thomas Friedl, Dept. EPSAG, Göttingen University for providing infrastructural support and the culture collections SAG and UTEX for supporting the scientific community on laboratory studies of microorganisms.

## Author contributions

A.H., J.d.V. and Y.Y. secured funding. M.L., J.M.A., J.d.V. and Y.Y. provided resources and materials. A.H., J.d.V. and Y.Y. provided supervision. J.d.V. and Y.Y. conceptualized the study. J.d.V., K.B. and Y.Y. performed project administration. X.F., E.F., W.S.G., T.D. and J.M.R.F.-J. performed experimental work, including algal culturing, DNA and RNA extraction and microscopy. A.H., C.P. and K.v.s. performed transmission electron microscopy. J.Z., X.F. and Y.Y. generated and annotated the draft genomes and transcriptomes. I.I., S.d.V. and J.d.V. analyzed phenylpropanoid metabolism-related genes. R.D.H. and I.V.G. coordinated genomes deposition/annotation in PhycoCosm. L.B. performed and evaluated Illumina assemblies. K.B. and C.D. coordinated Illumina sequencing for UTEX 1559 and UTEX 1560. L.G., F.R. and G.T. annotated and analyzed MADS-box genes. I.I., J.M.S.Z., T.P.R., A.D.A., A.D., A.M. and P.-M.D. analyzed phytohormone-related genes. J.B.A., N.K. and A.M. performed ABA measurements. H.B. analyzed data on cell division. F.W.-L. analyzed photoreceptor genes. K.v.s. helped in the initial phase of the project in strain purifications and mating experiments. J.K. and P.-M.D. performed phylogenetic analyses of symbiotic genes. M.J.B. and A.D. analyzed RNA-seq data. X.F., J.Z., B.Z., T.L., O.N., I.I., J.d.V. and Y.Y. analyzed data and generated data figures and tables. J.Z., X.W., N.F.-P., S.A.R. and Y.Y. conducted the WGD analysis. S.A.R. and R.P. performed TAPscan analysis. S.A.R. and F.H. performed contamination analyses. N.R. and C.P. established the protocols and performed the chromosome stainings, staining. X.F. and J.P.M. performed the organellar genome assembly and analysis. X.F., J.Z., B.Z., J.H. and Y.Y. performed the cell wall and HGT analysis. H.Y. and C.Z. performed the microexon analysis. Z.A. and M.M. built the *Zygnema* gene co-expression database. X.F., I.I., J.d.V. and Y.Y. performed comparative genomic analyses. X.F., I.I., J.Z., J.d.V. and Y.Y. wrote the original draft. All authors helped discuss the results and write the paper. X.F., J.Z. and I.I. contributed equally. J.d.V. and Y.Y. contributed equally.

## Competing interests

The authors declare no competing interests.

## Additional information

**Correspondence and requests for materials** should be addressed to Jan de Vries or Yanbin Yin.

# Reporting Summary

## Statistics

For all statistical analyses, confirm that the following items are present in the figure legend, table legend, main text, or Methods section.

| n/a | Confirmed | |
|---|---|---|
| ☐ | ☒ | The exact sample size (*n*) for each experimental group/condition, given as a discrete number and unit of measurement |
| ☐ | ☒ | A statement on whether measurements were taken from distinct samples or whether the same sample was measured repeatedly |
| ☒ | ☐ | The statistical test(s) used AND whether they are one- or two-sided<br>*Only common tests should be described solely by name; describe more complex techniques in the Methods section.* |
| ☐ | ☒ | A description of all covariates tested |
| ☐ | ☒ | A description of any assumptions or corrections, such as tests of normality and adjustment for multiple comparisons |
| ☒ | ☐ | A full description of the statistical parameters including central tendency (e.g. means) or other basic estimates (e.g. regression coefficient) AND variation (e.g. standard deviation) or associated estimates of uncertainty (e.g. confidence intervals) |
| ☒ | ☐ | For null hypothesis testing, the test statistic (e.g. $F$, $t$, $r$) with confidence intervals, effect sizes, degrees of freedom and $P$ value noted<br>*Give P values as exact values whenever suitable.* |
| ☒ | ☐ | For Bayesian analysis, information on the choice of priors and Markov chain Monte Carlo settings |
| ☒ | ☐ | For hierarchical and complex designs, identification of the appropriate level for tests and full reporting of outcomes |
| ☐ | ☒ | Estimates of effect sizes (e.g. Cohen's *d*, Pearson's *r*), indicating how they were calculated |

*Our web collection on statistics for biologists contains articles on many of the points above.*

## Software and code

Policy information about availability of computer code

| Data collection | Transmission electron micrographs were taken on a Zeiss Libra 120 transmission electron microscope (Carl Zeiss AG, Oberkochen, Germany) at 80 kV, which was equipped with a TRS 2k SSCCD camera and operated by ImageSP software (Albert Tröndle Restlichtverstärker Systeme, Moorenweis, Germany).<br><br>DNA extraction: Quality and quantity of purified DNA was evaluated by using 1% agarose gel electrophoresis, NanoDrop 2000/2000c Spectrophotometers, and Qubit 3.0 Fluorometer (Thermo Fisher Scientific).<br><br>Library preparation and sequencing: DNA samples were sequenced at Roy J. Carver Biotechnology Center at University of Illinois at Urbana-Champaign, using Oxford Nanopore and Illumina technologies (Table S1A). Oxford Nanopore DNA libraries were prepared with 1D library kit SQK-LSK109 and sequenced with SpotON R9.4.1 FLO-MIN106 flowcells for 48h on a GridIONx5 sequencer. Base calling was performed with Guppy v1.5 (https://community.nanoporetech.com). Illumina shotgun genomic libraries were prepared with the Hyper Library construction kit (Kapa Biosystems, Roche). Libraries had an average fragment size of 450 bp, from 250 to 500 bp, and sequenced with 2x250 bp paired-end on HiSeq 2500. Additional DNA samples were sequenced at the Genome Research Core in University of Illinois at Chicago and JGI. The Illumina shotgun genomic libraries were prepared with Nextera DNA Flex Library Prep Kit. The libraries had an average fragment size of 403 bp and sequenced with 2x150 bp paired-end on HiSeq 4000 (Table S1A). RNA samples were sequenced at the Genome Research Core in University of Illinois at Chicago. The libraries were prepared by rRNA depletion with Illumina Stranded Total RNA kit plus Ribo-Zero Plant (https://www.illumina.com/products/by-type/sequencing-kits/library-prep-kits/truseq-stranded-total-rna-plant.html), and 2x150 bp paired-end sequencing was performed on HiSeq 4000. RNA from the third batch of stress experiments were sequenced at the NGS-Integrative Genomics Core Unit of the University Medical Center Göttingen, Germany. Stranded mRNA libraries were prepared with the Illumina stranded mRNA kit and paired-end sequencing of 2x150 bp reads was carried out on an Illumina HiSeq 4000 platform. |
|---|---|

LC-MS/MS analysis of abscisic acid
Abscisic acid was determined in samples using an LC-MS/MS system which consisted of Nexera X2 UPLC (Shimadzu) coupled QTRAP 6500+ mass spectrometer (Sciex). Chromatographic separations were carried out using the Acclaim RSLC C18 column (150×2.1 mm, 2.2μm, Thermo Scientific) employing acetonitrile/water+0.1% acetic acid linear gradient. The mass spectrometer was operated in negative ESI mode. Data was acquired in MRM mode using following transitions: 1) ABA 263.2->153.1 (-14 eV), 263.2->219.1 (-18 eV); 2) ABA -D6 (IS) 269.2->159.1 (-14 eV), 269.2->225.1 (-18 eV); declustering potential was -45 V. Freeze-dried moss samples were ground using the metal beads in homogenizer (Bioprep-24) to a fine powder. Accurately weighted (about 20 mg) samples were spiked with isotopically labeled ABA -D6 (total added amount was 2 ng) and extracted with 1.5 ml acetonitrile/water (1:1) solution acidified with 0.1% formic acid. Extraction was assisted by sonication (Elma S 40 H, 15 min, two cycles) and solution was left overnight for completion of extraction. Liquid was filtered through 0.2 μm regenerated cellulose membrane filters, evaporated to dryness upon a stream of dry nitrogen and redissolved in 100 μl extraction solution.

Data analysis

Transcriptome assembly
Raw RNA-seq reads (Table S1A) were quality checked with FastQC v.0.11.9 (http://www.bioinformatics.babraham.ac.uk/projects/fastqc/) (Andrews 2010), trimmed with TrimGalore (https://github.com/FelixKrueger/TrimGalore), and were inspected again with FastQC. All reads were combined, and de novo assembled with Trinity version 2.9.0 (Grabherr et al. 2011; Haas et al. 2013).

K-mer frequency analysis
The trimmed DNA Illumina reads were filtered out with BLASTP version 2.13.0+ using plastomes and mitogenomes from Zygnema as references. Remaining (putatively nuclear) were used to predict the best k-mer size by kmergenie v1.7048 (http://kmergenie.bx.psu.edu/) (Chikhi and Medvedev 2014). The histogram of the best k-mer was then uploaded to GenomeScope for viewing the genome plot (http://qb.cshl.edu/genomescope/) (Vurture et al. 2017) (Table S1B and Figure S2).

Genome assembly and scaffolding
To assemble the genome of SAG 698-1b, a total of 5.4 Gb (82x) of Oxford Nanopore nuclei DNA reads were assembled with wtdbg2 (Ruan and Li 2020) (https://github.com/ruanjue/wtdbg). Assembled contigs were polished by Racon v1.4 (Vaser et al. 2017) and three iterations of pilon version 1.2 (Walker et al. 2014) with Illumina paired-end reads. The polished genome was scaffolded by Dovetail Genomics HiRise software with Hi-C sequencing data (https://dovetailgenomics.com/). Genome contamination was examined by BLASTX against NCBI's NR database and contaminated scaffolds were removed.
To assemble the UTEX 1559 genome, an initial assembly was done with SPAdes v3 (Antipov et al. 2016) using Illumina paired-end reads (2x150 bp), three mate-pair libraries (insert size: 3-5 kb; 5-7 kb and 8-10 kb) and Oxford Nanopore reads (Table S1A). Assembled contigs were further scaffolded by two rounds of Platanus-allee (Kajitani et al. 2019) with Illumina paired-end reads (2x 250 bp), three mate-pair libraries (insert size 3-5 kb; 5-7 kb and 8-10 kb) and Oxford Nanopore reads. For the UTEX 1560 genome, Illumina paired-end (2x150 bp) and PacBio HiFi reads were used for assembly with SPAdes and further scaffolded with Platanus-allee. Scaffolds with contaminations were identified by BLASTX against NR and removed. The genomes of UTEX1559 and UTEX1560 were scaffolded by Dovetail Genomics HiRise software with Hi-C sequencing data from SAG 698-1b.
The genome of SAG 698-1a_XF was sequenced with PacBio HiFi long reads (40 Gb), Nanopore long reads (4 Gb), and Illumina short reads (>100 Gb). The k-mer analysis using Illumina reads revealed two peaks in the k-mer distribution, suggesting that SAG 698-1a_XF exists as a diploid organism with an estimated heterozygosity rate of 2.22% (Figure S2). All Illumina short reads and the Nanopore reads were first assembled into contigs using SPAdes. Then, WENGAN (Di Genova et al. 2021) was used to assemble HiFi long reads and Illumina paired-end reads (2x150 bp) using the SPAdes contigs as the reference. Lastly, the resulting WENGAN v0.2 contigs were scaffolded and gaps were closed with Platanus-allee using all the Nanopore, HiFi, and Illumina reads to derive a consensus pseudo-haploid genome.
To evaluate the quality of assembled genomes (Table S1D), raw RNA-seq reads, Oxford Nanopore and Illumina DNA reads were mapped to the assembly with hisat v2 (Kim et al. 2019), minimap v2 (Li 2018), and bowtie v2 (Langmead and Salzberg 2012), respectively. To assess genome completeness, a BUSCO v.5.2.2 (Seppey et al. 2019) analysis was performed with the 'Eukaryota odb10' and 'Viridiplantae odb10' reference sets.

Repeat annotation
Repetitive DNA was annotated using the homology strategy with repeat libraries generated with RepeatModeler (v.2.0.1). RepeatModeler integrates RepeatScout, RECON, LTRharvest and LTR_retriever tools (version 2.0.1; http://www.repeatmasker.org/RepeatModeler/) (Flynn et al. 2020). The MITE (Miniature inverted-repeat transposable elements) library was identified with MITE-tracker (2018 release) (Crescente et al. 2018) software. These two identified libraries were combined and incorporated into RepeatMasker (version 4.0.9; http://www.repeatmasker.org/) for repeat annotation.

Genome annotation
In all four genomes, protein coding genes were predicted by the MAKER-P pipeline (v3.01.03) (Campbell et al. 2014) which integrates multiple gene prediction resources, including ab initio prediction, protein homology-based gene prediction and transcripts-based evidence. First, repetitive elements were masked by RepeatMasker with a custom repeat library generated by RepeatModeler. Rfam with infernal and tRNA-Scan2 were used to analyze non-coding RNA and tRNA. For the transcripts evidence, total of 103,967 transcripts were assembled by Trinity v2.9.0 (reference-free) and StringTie v2.1 (reference-based) with RNA-seq reads. Transcriptome assembly was used for generating a complete protein-coding gene models using PASA. Proteins from M. endlicherianum, S. muscicola and A. thaliana (TAIR10) were used as homology-based evidence. Then, the resulting protein-coding gene models from the first iteration of MAKER-P pipeline were used as training date set for SNAP and Augustus models, which were fed into MAKER for the second iteration of annotation. After three rounds of gene prediction, MAKER-P combined all the protein-coding genes as the final annotated gene sets.

Plastome and mitogenome assembly and annotation
NOVOPlasty 3.8.2 (https://github.com/ndierckx/NOVOPlasty) (Dierckxsens et al. 2017) was used to assemble plastomes. The contiguity of assembled plastomes was examined in Geneious software (https://www.geneious.com/) (Kearse et al. 2012) with read mapping. For SAG 698-1b mitogenome assembly, Oxford Nanopore reads were assembled with Canu (Koren et al. 2017), where one long mitogenome contig of 238,378 bp was assembled. This contig was circularized and polished with three rounds of pilon (Walker et al. 2014), that was further corrected with Illumina raw reads and compared with mitogenome of UTEX 1559 (MT040698; (Orton et al. 2020)) in Geneious. For SAG 698-1a_XF, PacBio HiFi reads were used for the assembly of its mitogenome.
Plastome and mitogenome annotation was performed with GeSeq (Tillich et al. 2017; v2021) (https://chlorobox.mpimp-golm.mpg.de/geseq.html). For plastome annotation, BLAT search and HMMER profile search (Embryophyta chloroplast) were used for coding sequence, rRNA and tRNA prediction; ARAGORN v1.2.38, ARWEN v1.2.3 and tRNAscan-SE v2.0.5 were used for tRNA annotation. For mitogenome annotation, Viridiplantae was use for BLAT Reference Sequences. The annotated gff files were uploaded for drawing circular organelle

genome maps on OGDRAW (https://chlorobox.mpimp-golm.mpg.de/OGDraw.html) (Greiner et al. 2019).

Comparison of Z. circumcarinatum genomes (SAG 698-1b, UTEX 1559, UTEX 1560)
Two approaches were used to compare the three genomes (Figure S8). The first approach was based on the whole genome alignment (WGA) by using MUMMER v4.0.0. The parameters "--maxmatch -c 90 -l 40" were set to align the three genomes and then "-i 90 -l 1000" were set to filter out the smaller fragments. The second approach focused on the gene content comparisons. Orthofinder was used to obtain ortholog groups (orthogroups) from genomes' annotated proteins. Orthofinder results led to a Venn diagram with unique genes (orthogroups with genes from only 1 genome), cloud genes (orthogroups with genes from only 2 genomes), and core genes (orthogroups with genes from all 3 genomes), which collectively form the pan-genome. Orthofinder could have failed to detect homology between very rapidly evolved orthologous genes, which leads to an under-estimation of core genes. Also, gene prediction may have missed genes in one genome but found them in other genomes. To address these issues, the raw DNA reads of each genome were mapped to the unique genes and cloud genes using BWA. This step was able to push more unique genes and cloud genes to core genes or push some unique genes to cloud genes. The following criteria were used to determine if an orthogroup needed to be re-assigned: (i) the number of reads and coverage calculated by bedtools are >10 and >0.8 for a gene, respectively, and (ii) >60% coverage of genes in the orthogroup find sequencing reads from the other genomes. After this step the final Venn diagram was made (Figure S8F), showing the counts of the final core genes, cloud genes, and unique genes.

Whole genome duplication (WGD) analysis
To identify possible WGDs, Ks and 4dtv values were calculated for each genome. First, all paralog pairs were identified using RBBH (Reciprocal Best BLAST Hit) method using protein sequences (E-value < 1e-6), following the method described by Bowman et al. (Bowman et al. 2017). RBBH paralog pairs were aligned with MAFFT v7.3.10 (Katoh and Standley 2013) and the corresponding nucleotide alignments were generated. Using RBBH alignments of paralog pairs, KaKs_Calculator2.0 (Wang et al. 2010) with the YN model and the calculate_4DTV_correction.pl script were run to calculate Ks and 4dtv values for each alignment, respectively. Ks = 0 and 4dtv = 0 values were filtered. The Ks and 4dtv distributions were fitted with a gaussian kernel density model using the seaborn package. For the SAG 698-1b chromosome-level genome, MCscan (Wang et al. 2012) was run to identify syntenic block regions with default parameters.

Species phylogeny and divergence time analysis
Sixteen representative genomes were selected, including two chlorophytes (Volvox carteri, Chlamydomonas reinhardtii), seven Zygnematophyceae (Zygnema circumcarinatum SAG 698-1b, UTEX 1559, UTEX 1560, Z. cf. cylindricum SAG 698-1a_XF, Mesotaenium endlicherianum, Penium margaritaceum, Spirogloea muscicola), four additional streptophyte algae (Chara braunii, Klebsormidium nitens, Chlorokybus melkonianii, Mesostigma viride), two bryophytes (Marchantia polymorpha, Physcomitrium patens) and a vascular plant (Arabidopsis thaliana). Orthogroups were generated by OrthoFinder version 2.5.2 (Emms and Kelly 2019) and 493 low-copy orthogroups containing ≤ 3 gene copies per genome were aligned with MAFFT v7.310 (Katoh and Standley 2013). Gene alignments were concatenated and gaps were removed by Gblocks version 0.91b (Castresana 2000). Phylogenetic tree was built using RAxML v.8 (Stamatakis 2014) with the "-f a" method and the PROTGAMMAJTT model, and support with 100 pseudoreplicates of non-parametric bootstrap. The tree was rooted on Chlorophyta.
Using the above methodology, additional phylogenetic analyses were performed with (i) the four Zygnema strains and (ii) the seven Zygnematophyceae genomes, in order to obtain a higher number of single-copy loci, 5,042 and 204, respectively (Figure S7).
Divergence time estimation was carried out with MCMCTree implemented in the PAML package version 4.10.0j (Yang 2007). The 493 low-copy orthogroup protein sequence alignment was converted to the corresponding nucleotide alignment for MCMCTree, in which 10 MCMC (Markov Chain Monte Carlo) chains were run, each for 1,000,000 generations (Table S1F). Three calibration were set in the reference tree according to Morris et al., (Morris et al. 2018) on the nodes Viridiplantae (972.4~669.9 Ma), Streptophyta (890.9~629.1 Ma) and Embrophyta (514.8~473.5 Ma).

Comparative genomics analysis
Sixteen representative genomes were selected, including two chlorophytes (Volvox carteri, Chlamydomonas reinhardtii), seven Zygnematophyceae (Zygnema circumcarinatum SAG 698-1b, UTEX 1559, UTEX 1560, Z. cf. cylindricum SAG 698-1a_XF, Mesotaenium endlicherianum, Penium margaritaceum, Spirogloea muscicola), four additional streptophyte algae (Chara braunii, Klebsormidium nitens, Chlorokybus melkonianii, Mesostigma viride), two bryophytes (Marchantia polymorpha, Physcomitrium patens) and a vascular plant (Arabidopsis thaliana).
Orthogroups were inferred with Orthofinder. Time-calibrated species phylogeny was built with low-copy orthogroups (≤ 3 gene copies). Divergence time estimation was carried out with MCMCTree (version from 2017). Expanded and contracted gene families were identified with CAFE and the species phylogeny. For microexon analyses, MEPmodeler(Yu et al. 2022) was used (https://github.com/yuhuihui2011/MEPmodeler).
For comparative genomics studies of multicellularity, the sixteen genomes were classified into two groups, the unicellular group (C. reinhardtii, C. melkonianii, M. viride, S. muscicola, M. endlicherianum, P. margaritaceum) and the multicellular group (V. carteri, K. nitens, C. braunii, SAG 698-1a_XF, SAG 698-1b, UTEX 1559, UTEX 1560, and M. polymorpha, P. patens, A. thaliana). Proteins in the 16 genomes were annotated by Pfam to find functional domains. Domain occurrences (presense/absence) and abundances in each genome were recorded, and were compared between the two groups of genomes to infer domain gain, loss, and combination.

Gene family phylogenetic analysis
CAZyme families were identified with dbCAN v2 (Zhang et al. 2018) with default parameters (E-value < 1e-10 and coverage > 0.35). Whenever needed, dbCAN2 was rerun by using more relaxed parameters. The experimentally characterized cell wall enzymes were manually curated from the literature (Data S1 and Table S1L). Reference genes were included into the phylogenies to infer the presence of orthologs across the 16 genomes and guide the split of large families into subfamilies. Phylogenetic trees were built by using FastTree initially, and for some selected families, RAxML (Stamatakis 2014) and IQ-Tree v1.5.5 (Nguyen et al. 2015) were used to rebuild phylogenies to verify topologies.

Orthogroup expansion and contraction analysis
We inferred expanded and contracted gene families with CAFE v.5 using orthogroups inferred with Orthofinder v.2.4.0 and the previously inferred time-calibrated species phylogeny. CAFE v.5 was run with default settings (base) using the inferred orthogroups and a calibrated species phylogeny. Two independent runs arrived to the same final likelihood and lambda values. The first eight orthogroups (OG0-7) were excluded from the analysis due to too drastic size changes between branches that hampered likelihood calculation; excluded orthogroups were mostly exclusive to a single Zygnema or Chara genome and likely represented transposable elements, as judged by results of BLASTP against NR.

Phytohormones

Proteins involved in phytohormone biosynthesis and signaling were identified by BLASTP against annotated proteomes (e-value<1e-6) using Arabidopsis genes as queries. For genes with ubiquitous domains (e.g. CIPK, CPK3, SNRK2, CDG1, BAK1), hits were filtered by requiring BLASTP coverage ≥50% of the query. Significant hits were then aligned (MAFFT auto) and maximum likelihood gene trees were inferred in IQ-Tree using best-fit models and 1000 replicates of SH-like aLRT branch support ('-m TEST -msub nuclear -alrt 1000'). The final sets of homologs were identified by visually inspecting gene trees and identifying the most taxonomically diverse clade (with high support of SH-aLRT>0.85) that included the characterized Arabidopsis proteins. Bubble plot was generated with ggplot2 in R.

Constructing the co-expression network and establishing the Zygnema database.
The Highest Reciprocal Rank (HRR) co-expression network for Z. circumcarinatum SAG 698-1b was built from all samples (19 growth conditions) and the Zygnema database was established using the CoNekT framework (Proost and Mutwil 2018). The gene co-expression clusters were identified using the Heuristic Cluster Chiseling Algorithm (HCCA) with standard settings (Mutwil et al. 2010).

Screening for symbiotic genes and phylogenetic analysis
Symbiotic genes were screened against a database of 211 plant species across Viridiplantae lineage using proteins of the model plant Medicago truncatula as queries in BLASTP v2.11.0+ (Camacho et al. 2009) searches with default parameters and an e-value < 1e-10. Initial alignments of all identified homologs was performed using the DECIPHER package (Wright 2015) in R v4.1.2 (R Core Team). Positions with >60% gaps were removed with trimAl v1.4 (Capella-Gutiérrez et al. 2009) and a phylogenetic analysis was performed with FastTree v2.1.11 (Price et al. 2009). Clades corresponding to M. truncatula orthologs queries were extracted and a second phylogeny was performed. Proteins were aligned using MUSCLE v3.8.1551 (Edgar 2004) with default parameters and alignments cleaned as described above. Tree reconstruction was performed using IQ-Tree v2.1.2 (Minh et al. 2020) based on BIC-selected model determined by ModelFinder (Kalyaanamoorthy et al. 2017). Branch supports was estimated with 10,000 replicates each of both SH-aLRT (Guindon et al. 2010) and UltraFast Bootstraps (Hoang et al. 2018). Trees were visualized and annotated with iTOL v6 (Letunic and Bork 2021). For the GRAS family, a subset of 42 species representing the main lineages of Viridiplantae was selected and GRAS putative proteins screened using the HMMSEARCH program with default parameters and the PFAM domain PF03514 from HMMER3.3 (Johnson et al. 2010) package. Phylogenetic analysis was then conducted as described above.

Screening for CCD homologs and phylogenetic analysis
Annotated proteins from 21 land plant genomes (Amborella trichopoda, Anthoceros agrestis, Anthoceros punctatus, Arabidopsis lyrata, Arabidopsis thaliana, Azolla filiculoides, Brachypodium distachyon, Brassica rapa, Lotus japonicus, Marchantia polymorpha, Medicago truncatula, Oryza sativa, Physcomitrium patens, Picea abies, Pisum sativum, Salvinia cucullata, Selaginella moellendorffii, Sphagnum fallax, Spinacia oleracea, Gnetum montanum, Crocus sativus), 7 streptophyte algal genomes (Spirogloea muscicola, Penium_margaritaceum, Mesotaenium endlicherianum, Mesostigma viride, Klebsormidium nitens, Chlorokybus melkonianii, Chara braunii, Zygnema circumcarinatum), 6 chlorophyte genomes (Ulva mutabilis, Ostreococcus lucimarinus, Micromonas pusilla, Micromonas sp., Chlamydomonas reinhardtii, Coccomyxa subellipsoidea, Chlorella_variabilis), 5 cyanobacterial genomes (Trichormus azollae, Oscillatoria acuminata, Nostoc punctiforme, Gloeomargarita lithophora, Fischerella thermalis), as well as the transcriptome of Coleochaete scutata (de Vries et al. 2018). The representative A. thaliana protein was used as query for BLASTP searches against the above annotated proteins (E-value < 1e-5). Homologs were aligned with MAFFTv7.453 L-INS-I approach (Katoh and Standley 2013) and maximum likelihood phylogenies computed with IQ-Tree v.1.5.5 (Nguyen et al. 2015), with 100 bootstrap replicates and BIC-selected model (WAG+R9) with ModelFinder (Kalyaanamoorthy et al. 2017). Functional residue analyses were done based on published structural analysis (Messing et al. 2010), and the alignments were plotted with ETE3 (Huerta-Cepas et al. 2016).

Phylogeny of MADS-box genes
MADS-domain proteins were identified by Hidden Markov Model (HMM) searches (Eddy 1998) on annotated protein collections. MADS-domain sequences of land plants and opisthokonts were taken from previous publications (Marchant et al. 2022; Gramzow et al. 2010). MADS domain proteins of other streptophyte algae were obtained from the corresponding genome annotations and transcriptomic data (One Thousand Plant Transcriptomes Initiative, 2019) (One thousand plant transcriptomes and the phylogenomics of green plants 2019). Additional MADS-domain proteins of Zygnematophyceae were identified by BLAST against transcriptome data available at NCBI's sequence read archive (SRA) (Sayers et al. 2021). MADS-domain-protein sequences were aligned using MAFFTv7.310 (Katoh and Standley 2013) with default options. Sequences with bad fit to the MADS domain were excluded and the remaining sequences realigned, and trimmed using trimAl (Capella-Gutiérrez et al. 2009) with options "-gt .9 -st .0001". A maximum likelihood phylogeny was reconstructed using RAxMLv8.2.12 (Stamatakis 2014) on the CIPRES Science Gateway (Miller et al. 2011).

For manuscripts utilizing custom algorithms or software that are central to the research but not yet described in published literature, software must be made available to editors and reviewers. We strongly encourage code deposition in a community repository (e.g. GitHub). See the Nature Portfolio guidelines for submitting code & software for further information.

# Data

Policy information about availability of data

All manuscripts must include a data availability statement. This statement should provide the following information, where applicable:

- Accession codes, unique identifiers, or web links for publicly available datasets
- A description of any restrictions on data availability
- For clinical datasets or third party data, please ensure that the statement adheres to our policy

Data and code availability
The four Zygnema genomes, raw DNA reads, and rRNA-depleted RNA-seq of SAG 698-1b can be accessed through NCBI BioProject PRJNA917633. The raw DNA read data of UTEX1559 and UTEX1560 sequenced by the Joint Genome Institute can be accessed through BioProjects PRJNA566554 and PRJNA519006, respectively. RNA-seq data of UTEX1559 can be accessed through BioProject PRJNA524229. Poly-A enriched RNA-seq data of SAG 698-1b can be accessed through BioProject PRJNA890248 and the Sequence Read Archive (SRA) under the accession SRR21891679 to SRR21891705. Zygnema genomes are also available through the PhycoCosm portal129 (https://phycocosm.jgi.doe.gov/SAG698-1a; https://phycocosm.jgi.doe.gov/SAG698-1b; https://phycocosm.jgi.doe.gov/UTEX1559; https://phycocosm.jgi.doe.gov/UTEX1560). Data files are available at Figshare https://doi.org/10.6084/m9.figshare.22568197 and Mendeley under doi: 10.17632/gk965cbjp9.1
No original code was used; all computational analyses were performed with published tools and are cited in the Methods section.

# Research involving human participants, their data, or biological material

Policy information about studies with [human participants or human data](). See also policy information about [sex, gender (identity/presentation), and sexual orientation]() and [race, ethnicity and racism]().

| | |
|---|---|
| Reporting on sex and gender | n/a |
| Reporting on race, ethnicity, or other socially relevant groupings | n/a |
| Population characteristics | n/a |
| Recruitment | n/a |
| Ethics oversight | n/a |

Note that full information on the approval of the study protocol must also be provided in the manuscript.

# Field-specific reporting

Please select the one below that is the best fit for your research. If you are not sure, read the appropriate sections before making your selection.

☒ Life sciences ☐ Behavioural & social sciences ☐ Ecological, evolutionary & environmental sciences

For a reference copy of the document with all sections, see [nature.com/documents/nr-reporting-summary-flat.pdf]()

# Life sciences study design

All studies must disclose on these points even when the disclosure is negative.

| | |
|---|---|
| Sample size | Analyses were done on millions of pooled filaments of Zygnema (cultures were grown to a density on plate during which healthy growth still occurred but the whole plate was covered to yield appropriate biomass for extraction of nucleic acids). Sequencing was performed to a depth that was choes based on approaching saturation level (based on obtaining global gene expression patterns given the number of genes in the genomes of Zygnema). For chromosome counting, a minimum of 10 cells each from three independent cell cultures were analyzed. TEM was performed using two independent cell cultures and each time ≥15 algal filaments. |
| Data exclusions | No data were excluded |
| Replication | RNAseq under different conditions was done on at least three independent biological replicates per condition, all of which were used for this study. All attempts at replication were successful. For chromosome counting, three independent cell cultures were analyzed. |
| Randomization | Apart from taking a random selection of millions of filaments from a liquid culture, samples were not additionally randomized selected for any experiment. Treatments were designed so that no batch effect due to setups occurred (defined media, exact concentrations of challenges, monitoring of growth light and temperature etc.) |
| Blinding | Blinding was not relevant for this study. It is irrelevant for the genome and transcriptome analyses because we always worked with all data, using all versus all comparisons, unsupervised methods, and fully transparent pipelines. All phenotypic evaluation (e.g. counts of chromosomes) are quantifyable and unambigous. |

# Reporting for specific materials, systems and methods

We require information from authors about some types of materials, experimental systems and methods used in many studies. Here, indicate whether each material, system or method listed is relevant to your study. If you are not sure if a list item applies to your research, read the appropriate section before selecting a response.

## Materials & experimental systems

| n/a | Involved in the study |
|---|---|
| ☒ | ☐ Antibodies |
| ☐ | ☒ Eukaryotic cell lines |
| ☒ | ☐ Palaeontology and archaeology |
| ☒ | ☐ Animals and other organisms |
| ☒ | ☐ Clinical data |
| ☒ | ☐ Dual use research of concern |
| ☐ | ☒ Plants |

## Methods

| n/a | Involved in the study |
|---|---|
| ☒ | ☐ ChIP-seq |
| ☒ | ☐ Flow cytometry |
| ☒ | ☐ MRI-based neuroimaging |

## Eukaryotic cell lines

Policy information about cell lines and Sex and Gender in Research

| Cell line source(s) | Z. circumcarinatum SAG 698-1b and Z. cf. cylindricum SAG 698-1a were obtained from the Culture Collection of Algae at Göttingen University (SAG) (https://sagdb.uni-goettingen.de). Z. circumcarinatum UTEX 1559 and UTEX 1560 were obtained from the UTEX Culture Collection of Algae at UT-Austin (https://utex.org/). From Z. cf. cylindricum SAG 698-1a, a single filament was picked and a new culture was established, coined Z. cf. cylindricum SAG 698-1a_XF; Z. cf. cylindricum SAG 698-1a_XF was used for genome sequencing. |
|---|---|
| Authentication | Authentication was carried out by the Experimental Phycology and Culture Collection of Algae in Göttingen, Germany, via microscopy and genetic markers. |
| Mycoplasma contamination | n/a |
| Commonly misidentified lines (See ICLAC register) | n/a |

## Dual use research of concern

Policy information about dual use research of concern

### Hazards

Could the accidental, deliberate or reckless misuse of agents or technologies generated in the work, or the application of information presented in the manuscript, pose a threat to:

| No | Yes |
|---|---|
| ☒ | ☐ Public health |
| ☒ | ☐ National security |
| ☒ | ☐ Crops and/or livestock |
| ☒ | ☐ Ecosystems |
| ☒ | ☐ Any other significant area |

### Experiments of concern

Does the work involve any of these experiments of concern:

| No | Yes |
|---|---|
| ☒ | ☐ Demonstrate how to render a vaccine ineffective |
| ☒ | ☐ Confer resistance to therapeutically useful antibiotics or antiviral agents |
| ☒ | ☐ Enhance the virulence of a pathogen or render a nonpathogen virulent |
| ☒ | ☐ Increase transmissibility of a pathogen |
| ☒ | ☐ Alter the host range of a pathogen |
| ☒ | ☐ Enable evasion of diagnostic/detection modalities |
| ☒ | ☐ Enable the weaponization of a biological agent or toxin |
| ☒ | ☐ Any other potentially harmful combination of experiments and agents |

