## [Peer Review File · Nature Genetics]

Peer Review Information

Manuscript Title: Genomes of multicellular algal sisters to land plants illuminate signaling network evolution

Corresponding author name(s): Professor Jan de Vries, Dr Yanbin Yin

Editorial Notes:

Transferred manuscripts This document only contains reviewer comments, rebuttal and decision letters for versions considered at Nature Genetics.

Reviewer Comments & Decisions:

Decision Letter, initial version:

21st Jun 2023

Dear Professor de Vries,

Your Article, "Genomes of multicellular algal sisters to land plants illuminate signaling network evolution" has now been seen by 2 referees. You will see from their comments below that while they find your work of interest, some important points are raised. We are interested in the possibility of publishing your study in Nature Genetics, but would like to consider your response to these concerns in the form of a revised manuscript before we make a final decision on publication.

To guide the scope of the revisions, the editors discuss the referee reports in detail within the team with a view to identifying key priorities that should be addressed in revision. In this case, we think both referees have provided constructive reviews aimed at strengthening the analyses and improving the presentation. We particularly ask that you address their comments as thoroughly as possible with appropriate revisions. We hope that you will find the prioritized set of referee points to be useful when revising your study.

We therefore invite you to revise your manuscript taking into account all reviewer and editor comments. Please highlight all changes in the manuscript text file. At this stage we will need you to upload a copy of the manuscript in MS Word .docx or similar editable format.

We are committed to providing a fair and constructive peer-review process. Do not hesitate to contact us if there are specific requests from the reviewers that you believe are technically impossible or

unlikely to yield a meaningful outcome.

*2) If you have not done so already please begin to revise your manuscript so that it conforms to our Article format instructions, available here. Refer also to any guidelines provided in this letter.

Please be aware of our guidelines on digital image standards.

[redacted]

We hope to receive your revised manuscript within 3 to 6 months. If you cannot send it within this time, please let us know.

Sincerely,
Wei

Wei Li, PhD
Senior Editor
Nature Genetics
New York, NY 10004, USA
www.nature.com/ng

Reviewers' Comments:

Reviewer #1:

Remarks to the Author:

Feng et al sequence 4 genomes of filamentous Zygnematophyceae, the multicellular algal sisters of land plants. Three of these genomes are the first chromosome-scale assemblies for any streptophyte algae. To date, only unicellular Zygnematophyceae have been sequenced. Inclusion of filamentous (multicellular) species is important for understanding the transition from water to land, or terrestrialization.

Despite nearly identical names SAG 198-1b and SAG-198-1a_XF for two of their species, corresponding genomes differed in size by a factor of 5 (the authors incorrectly said a factor of 4). There was a SAG-198-1a mentioned, which the authors implied was distinct from SAG-198-1a_XF, but I am not sure it's worth confusing the readers with such details (especially as it's never discussed again).

The authors presented the expected comparisons of gene content for Chlorophyta, Embryophyta, and Zygnematophyceae. Given that the primary claim to novelty in this manuscript is that they sequenced filamentous (multicellular) Zygnematophyceae, I was shocked that they did not discuss how their results compare to similar analyses in previous manuscripts for unicellular Zygnematophyceae OR break their results down accordingly. Some of this information can be inferred by a closer examination of the figures and supplemental data, but not all of it, and readers have every right to expect the authors to do it for them. For example, Figure 2a clearly shows a loss of many orthogroups in the filamentous (multicellular) species relative to their unicellular brethren. No mention is made of this, and the discussions make no effort to tell the readers what is observed in filamentous (multicellular) species versus unicellular versus both. I think this is an egregious omission that must be addressed in any future resubmissions.

Figure 4 on co-expression networks and the evolution of the plant perceptron is interesting, but most readers will need more introductory materials to appreciate what is being said. The networks they show are illegible in printed form (they require an interactive viewer to zoom in and out). More importantly, the authors must explain how these illegible networks support the plant perceptron concept in the 2017 Nature paper by Scheres and van der Putten. It is indeed a trendy concept, with everybody these days marvelling at the breakthroughs in deep learning AI (using gigantic perceptrons). Precisely because this is so trendy, they cannot just wave their hands and claim to have proven a connection.

The early origin in algae of genes once thought unique to land plants has been pointed out in other publications. No harm emphasizing the point though. Similarly, micro-exons have been reported before, but like the early origins idea, not yet widely known and therefore worth emphasizing. All-in-all, this study has produced enough material to be published eventually, but it has to be better written.

A number of points in the tables and figures need to be clarified.

Table 1: Explain the ** next to the UTEX1560 mapping rate.

Figure 2a: Explain the numbers adjacent to each of the OG circles.

Figure 2e: There is no obvious plot; just a mysterious arrow.

Figure 4bcd: Says 52 117 & 128 in image but 38 21 & 52 in caption.

Figure 4ef: Plots are missing or perhaps mislabeled as C & D.

Figure 5b: Explain the numbers (weirdly not proportional to sizes).

More generally, there were a lot of notations in their figures that were not explained. The authors must either document whatever is shown or don't show it at all.

Reviewer #2:

Remarks to the Author:

A. The paper presents the analysis and summary of the first genomes of the Zygnematophyceae. From a comparative genomics standing, they have further refined our understanding of process of terrestrialisation and the parallels of evolution between land plants and their algal relatives.

B. The results are novel, these are new genomes. The results fit the current narrative about the evolution of signalling pathways and complexity in green plants, but this is not a criticism of the paper.

C. As far as I can tell, the methods are sound and correct throughout and the presented results are interesting and well supported.

D. Statistics are fine.

E. The conclusions are supported.

F. Improvements... As ever, these papers tend to tease further interesting results that are yet to come, but I think that what is presented within is of sufficient interest.

I am interested by the independent expansion of the ABA pathway in Zygnema! I also wonder whether there are parallels in the evolution of multicellularity in Zygnema and land plants compared to the single cell zygnematophyceae?

I also wonder how this compares to the brief analysis of algae in Harris et al. (2022) NEE. They saw large amounts of gene loss leading to crown Zygnematophyceae - is this then being reversed by further gains in Zygnema?

These questions are of interest but mostly for my own curiosity!

G. The references are correct

H. It is clear to read.

I would recommend publication, with any minor edits at the editors discretion.

Author Rebuttal to Initial comments

Reviewers' Comments:

Reviewer #1:

Remarks to the Author:

Feng et al sequence 4 genomes of filamentous Zygnematophyceae, the multicellular algal sisters of land plants. Three of these genomes are the first chromosome-scale assemblies for any streptophyte algae. To date, only unicellular Zygnematophyceae have been sequenced. Inclusion of filamentous (multicellular) species is important for understanding the transition from water to land, or terrestrialization.

>>>>AU: We thank the reviewer for appreciating the scope and analyses that went into our work. Further, we would like to thank the reviewer for the input that stimulated some additional analyses, shedding light on the data from a different angle and thus, we think, making it more accessible.

Despite nearly identical names SAG 198-1b and SAG-198-1a_XF for two of their species, corresponding genomes differed in size by a factor of 5 (the authors incorrectly said a factor of 4). There was a SAG-198-1a mentioned, which the authors implied was distinct from SAG-198-1a_XF, but I am not sure it's worth confusing the readers with such details (especially as it's never discussed again).

>>>>AU: Thank you for bringing up this point, which we—being one first-name-basis with the strains—did not notice. Thank you. We made sure to fix this as follows: The nearly identical strain numbers SAG 698-1b and SAG-698-1a cannot be changed, since these are the official unique (and findable / needed for ordering) strain numbers. We now always use SAG-698-1a_XF in the text; the interested reader can still find the information on the relationship between SAG-698-1a_XF and SAG-698-1a in the methods section.

The authors presented the expected comparisons of gene content for Chlorophyta, Embryophyta, and Zygnematophyceae. Given that the primary claim to novelty in this manuscript is that they sequenced filamentous (multicellular) Zygnematophyceae, I was shocked that they did not discuss how their results compare to similar analyses in previous manuscripts for unicellular Zygnematophyceae OR break their results down accordingly. Some of this information can be inferred by a closer examination of the figures and supplemental data, but not all of it, and readers have every right to expect the authors to do it for them. For example, Figure 2a clearly shows a loss of many orthogroups in the filamentous (multicellular) species relative to their unicellular brethren. No mention is made of this, and the discussions make no effort to tell the readers what is observed in filamentous (multicellular) species versus unicellular versus both. I think this is an egregious omission that must be addressed in any future resubmissions.

>>>>AU: We thank the reviewer for raising this important point. Indeed, it was one of our main goals to provide insights into the evolution of filamentous growth. However, this is not as straightforward as one (or at least we) might initially think. This resulted in several lines of information that we previously presented briefly or outright removed (because of space restrictions), but which we now elaborate upon. We further performed a set of additional analyses (see below) that resulted

in an all-new Figure 3. All the information (old and new) we aimed to now consolidate, interconnect, and present in a clear manner in this revised version.

To first set the stage for understanding the type of multicellular growth in *Zygnema*, we have provided ultrastructural micrographs using transmission electron microscopy, highlighting the cell wall details that bear upon cell plate formation.

We made a completely new Figure 3 that focuses on comparisons between uni- and multicellular streptophytes—building on domain loss, gain, and combinations. We pinpoint a set of noteworthy domain combinations and genes. Yet, the gross numbers of unique domains are relatively low, which we now point out are "patterns align with proposed concepts on the evolution of multicellularity in green algae" (see work e.g., by James Umen). What appears more important is regulation, which is why we emphasize this point also in light of our co-expression networks (see also the answer to your query on the co-expression networks).

Our co-expression analyses, as highlighted in the initial submission, capture the interconnectivity between cell cycle, known cell division regulators, cell division effectors, and cell wall biosynthesis—speaking of plant growth and developmental genes that are at the heart of the molecular machinery that makes a multicellular organism. We now split the results and discussion section and highlight how all analyses, especially including the co-expression data, bear upon the emergence of genetic networks that underpin the molecular mechanism behind filamentous growth (trait actualization). We further stress that: "Throughout their evolutionary history, Zygnematophyceae have transition several times between multicellular and unicellular body plans (Hess et al., 2022). A parsimonious explanation is that a shared toolkit for multicellularity was present in the LCA of Zygnematophyceae, which comes to bear in filamentous genera but is still lingering as genetic potential in zygnematophyte unicells. And indeed, our data on shared orthogroups expansions recover several important regulatory genes for increasing cellular complexity for the LCAs of (i) Z+E and (ii) Zygnematophyceae. That said, we recover specific domain combinations that might underpin actualization of filamentous growth."

One last comment. There seems to be a misunderstanding from Figure 2a: only two orthogroups show significant contraction in the common filamentous ancestor of *Zygnema* spp. (elongation factor Tu and MLO, both were mentioned). The large set of contraction happened within the filamentous genus *Zygnema* spp., in the common ancestor of *Zygnema circumcarinatum*. These were also discussed — but of course not in the context of multicellularity, since this contraction is just a phenomenon within this filamentous genus (hence the trait multicellularity is not influenced by this contraction).

Figure 4 on co-expression networks and the evolution of the plant perceptron is interesting, but most readers will need more introductory materials to appreciate what is being said. The networks they show are illegible in printed form (they require an interactive viewer to zoom in and out). More importantly, the authors must explain how these illegible networks support the plant perceptron concept in the 2017 Nature paper by Scheres and van der Putten. It is indeed a trendy concept, with everybody these days marvelling at the breakthroughs in deep learning AI (using gigantic perceptrons). Precisely because this is so trendy, they cannot just wave their hands and claim to have proven a connection.

>>>>AU: We thank the reviewer about the feedback on the concept of the perceptron, which we deem an essential aspect of our study — not least because we think that tangoes with multicellularity, as explained above. This significance is attributed to its capacity to facilitate the linkage between environmental input and the consequential impact on growth and developmental processes, thereby yielding growth changes as an outcome and providing implicit directionality. The perceptron concept possesses the capacity to elucidate numerous facets of multicellular development in plants, contingent upon their inherent plasticity. In this context, the presence of a directed graph or explicit directionality within the network, in our estimation, does not constitute an imperative prerequisite. This assertion stems from the inherent clarity in the relationship between environmental perception and its influence on growth.

We agree with the reviewers point that this should be more clearly described. We thus placed emphasis on the likely flow of information. For this, we now properly introduce the layer of the perceptron, elaborate more in the text, and

pinpoint meaningful cohorts from each layer, and devoted more than half of the now re-structured discussion section to these aspects.

Specifically, we have worked with the genes in the module as puzzle pieces—and what they tell us based on the homology-based annotation. Since genes that are co-expressed are functionally related, they speak for concerted action in a genetic program. We recover programs from each layer, and connections between layers: We recover clear programs that are biologically meaningful, e.g. cohorts of growth & development / cell division / cell wall. These are clearly intrinsic genetic programs, covering the basal layer. We also recover genes that are for environmental sensing and signaling. These undoubtedly receive extrinsic input, providing implicit directionality, from outside/environment to inside. And we find them co-expressed with intrinsic growth programs. Hence, we illuminate parts that could work on connecting inside and outside.

We now better define cohorts of genes that, by their nature, must act in a genetic hierarchy (transduction through kinase cascades), transcription factors (there must be an upstream and downstream) in both the text and a re-worked Figure on the perceptron (now Fig. 5b). Thus, we point to the puzzle pieces that can make up directionality in the flow of information.

The early origin in algae of genes once thought unique to land plants has been pointed out in other publications. No harm emphasizing the point though. Similarly, micro-exons have been reported before, but like the early origins idea, not yet widely known and therefore worth emphasizing. All-in-all, this study has produced enough material to be published eventually, but it has to be better written.

>>>>AU: We thank the reviewer for the detailed comments. We highly appreciate this—it is apparent that the reviewer thought deeply about our study. Thank you!

A number of points in the tables and figures need to be clarified.

Table 1: Explain the ** next to the UTEX1560 mapping rate.

>>>>AU: We added the explanation. To calculate a robust mapping rate, and owing to the strain identity, we used the much larger volume of RNAseq data on SAG 698-1b.

Figure 2a: Explain the numbers adjacent to each of the OG circles.

>>>>AU: These are the numbers of OGs. We have added this to the figure legend.

Figure 2e: There is no obvious plot; just a mysterious arrow.

>>>>AU: Thank you, we did not notice that this could be confusing. We have swapped the label for 2e and the actual panel content of 2e to make it more obvious.

Figure 4bcd: Says 52 117 & 128 in image but 38 21 & 52 in caption.

>>>>AU: This part of the figure has been completely re-worked.

Figure 4ef: Plots are missing or perhaps mislabeled as C & D.

>>>>AU: As part of re-working the figure, we have re-labeled the panels.

Figure 5b: Explain the numbers (weirdly not proportional to sizes).

>>>>AU: The numbers are now explained in the figure legend.

More generally, there were a lot of notations in their figures that were not explained. The authors must either document whatever is shown or don't show it at all.

>>>>AU: Thank you. We have now included more information in all figure legends.

Reviewer #2:

Remarks to the Author:

A. The paper presents the analysis and summary of the first genomes of the Zygnematophyceae. From a comparative genomics standing, they have further refined our understanding of process of terrestrialisation and the parallels of evolution between land plants and their algal relatives.

>>>>AU: We thank the reviewer for appreciating the implications of our work.

B. The results are novel, these are new genomes. The results fit the current narrative about the evolution of signalling pathways and complexity in green plants, but this is not a criticism of the paper.

>>>>AU: Thank you.

C. As far as I can tell, the methods are sound and correct throughout and the presented results are interesting and well supported.

>>>>AU: Thank you.

D. Statistics are fine.

>>>>AU: Thank you.

E. The conclusions are supported.

>>>>AU: Thank you.

F. Improvements... As ever, these papers tend to tease further interesting results that are yet to come, but I think that what is presented within is of sufficient interest.

>>>>AU: Spot on. We have tried to highlight what we consider the most relevant and what we think will propel discussions in the field.

I am interested by the independent expansion of the ABA pathway in Zygnema! I also wonder whether there are parallels in the evolution of multicellularity in Zygnema and land plants compared to the single cell zygnematophyceae?

>>>>AU: This is an interesting point. To scrutinize this, we have computed a maximum likelihood phylogeny (JTT+F+R7 chosen according to Bayesian Information Criterion, 1000 Felsenstein bootstrap pseudo-replicates) of the expanded PP2Cs

(new Suppl. Figure S10B). We conclude that “the expansions of PP2Cs are shared among Zygnema spp. but independent of the radiation of PP2CAs in land plants”.

I also wonder how this compares to the brief analysis of algae in Harris et al. (2022) NEE. They saw large amounts of gene loss leading to crown Zygnematophyceae - is this then being reversed by further gains in Zygnema?

>>>>AU: This is an excellent point. Indeed, the dynamics of gain/loss and expansion/contraction within Zygnematophyceae is noteworthy: now that we have added more genomes, this inferred reduction appears to shrink. And we think this makes sense. After all, Zygnematophyceae are an extremely species-rich class of algae, with a within-clade 500-million-year divergence. We added this also to the discussion, where we highlight that “Our data indicate the dynamics in Zygnematophyceae genome evolution (Fig. 2a), highlighting the need for a phylo diverse comparative and complementing approach to understand the ancestor of land plants and algae.”

These questions are of interest but mostly for my own curiosity!

>>>>AU: We are glad that our paper sparked curiosity and interest! Much appreciated.

G. The references are correct

>>>>AU: Thank you.

H. It is clear to read.

>>>>AU: Thank you.

I would recommend publication, with any minor edits at the editors discretion.

>>>>AU: We thank the reviewer for the kind words and for appreciating the value of our study.

Decision Letter, first revision:

2nd Nov 2023

Dear Professor de Vries,

Your Article, "Genomes of multicellular algal sisters to land plants illuminate signaling network evolution" has now been seen by 2 referees. You will see from their comments below that while they find your work of interest, some important points are raised by Reviewer #1. We are interested in the possibility of publishing your study in Nature Genetics, but would like to consider your response to these concerns in the form of a revised manuscript before we make a final decision on publication.

We therefore invite you to revise your manuscript taking into account all reviewer comments. Please highlight all changes in the manuscript text file. At this stage we will need you to upload a copy of the manuscript in MS Word .docx or similar editable format.

We are committed to providing a fair and constructive peer-review process. Do not hesitate to contact

us if there are specific requests from the reviewers that you believe are technically impossible or unlikely to yield a meaningful outcome.

*2) If you have not done so already please begin to revise your manuscript so that it conforms to our Article format instructions, available here.

*3) Include a revised version of any required Reporting Summary:

Please be aware of our guidelines on digital image standards.

[redacted]

We hope to receive your revised manuscript within four to eight weeks. If you cannot send it within this time, please let us know.

Sincerely,

Wei

Wei Li, PhD
Senior Editor
Nature Genetics
New York, NY 10004, USA
www.nature.com/ng

Reviewers' Comments:

Reviewer #1:

Remarks to the Author:

Feng et al have made a good faith effort to address my comments. Aside from issues of clarity, there were two substantive issues.

Their claim to novelty is that they sequenced filamentous (multicellular) Zygnematophyceae, so it was imperative that there be a comparison to the published unicellular genomes for this taxa. A new section entitled "Multicellularity and protein domain gains, losses, and combinations" was provided. They did not however find any compelling changes in the protein domains that would explain multicellularity and deferred the explanation to the discussions. That being the case, I wonder if this section is longer than required, especially as I suspect they are already over their page limits.

Another section called "Co-expression networks, multicellular growth, and the evolution of the plant perceptron" tries to tie everything together. I have mixed feelings about the perceptron concept. As they say, "frequent overlap between receptors and transducers and between transducers and downstream targets suggest a hierarchy where environmental cues are received, transmitted and processed, allowing a complex downstream response that integrates a variety of extrinsic and intrinsic signals." To some, it might appear profound; but at least in my mind, it's obvious that something like this must be happening, even if the details are difficult to tease out. The word perceptron also triggers an association with the trendiest field in all of science, artificial intelligence (AI) by deep learning. Nothing in this paper makes a substantive contribution to AI – a point I may be overly sensitive to as I have a long familiarity with AI. Perhaps it's an unfortunate choice of words, dating back to the 2017 Nature paper from Scheres and van der Putten. Unless they believe there is a link to AI, in which case they have a LOT of explaining to do, they must disown this association.

Getting back to multicellularity, the discussions say, "a shared toolkit for multicellularity was present in the LCA of Zygnematophyceae, which comes to bear in filamentous genera but is still lingering as genetic potential in zygnematophyte unicells". This is certainly plausible, and I suspect it's true, although I can't say they proved it by not finding the requisite changes in protein domains.

Does this qualify for publication in a high-impact journal? Historically yes, because the data will be highly cited, even if the conclusions are tentative. As sequencing has gotten more routine, many authors feel compelled to hype their results. I understand and sympathize. But it's borderline dishonest. If they would tone it down, I would be satisfied.

Reviewer #2:

Remarks to the Author:

I have read the latest version of the manuscript and the responses of the authors. I am now happy to recommend the manuscript for publication, with my apologies for submitting my review later than intended.

Author Rebuttal, first revision:

Reviewers' Comments:

Reviewer #1:

Remarks to the Author:

Feng et al have made a good faith effort to address my comments. Aside from issues of clarity, there were two substantive issues.

>>>>AU: We thank the reviewer again for the helpful comments. We do think that by addressing these, our manuscript has improved.

Their claim to novelty is that they sequenced filamentous (multicellular) Zygnematophyceae, so it was imperative that there be a comparison to the published unicellular genomes for this taxa. A new section entitled “Multicellularity and protein domain gains, losses, and combinations” was provided. They did not however find any compelling changes in the protein domains that would explain multicellularity and deferred the explanation to the discussions. That being the case, I wonder if this section is longer than required, especially as I suspect they are already over their page limits.

>>>>AU: We have reduced the length of this section now by several hundred words by (i) shortening the text while keeping the same information/statements, and (ii) moving parts to the supplementary text.

Another section called “Co-expression networks, multicellular growth, and the evolution of the plant perceptron” tries to tie everything together. I have mixed feelings about the perceptron concept. As they say, “frequent overlap between receptors and transducers and between transducers and downstream targets suggest a hierarchy where environmental cues are received, transmitted and processed, allowing a complex downstream response that integrates a variety of extrinsic and intrinsic signals.” To some, it might appear profound; but at least in my mind, it’s obvious that something like this must be happening, even if the details are difficult to tease out. The word perceptron also triggers an association with the trendiest field in all of science, artificial intelligence (AI) by deep learning. Nothing in this paper makes a substantive contribution to AI – a point I may be overly sensitive to as I have a long familiarity with AI. Perhaps it’s an unfortunate choice of words, dating back to the 2017 Nature paper from Scheres and van der Putten. Unless they believe there is a link to AI, in which case they have a LOT of explaining to do, they must disown this association.

>>>>AU: This is an important comment as other readers might stumble across our use of words, too. As you point out, we do not — and never intended to — make any contribution / connection to AI. As you said, the word perceptron was not intentionally used for that reason, but simply because it was coined as the ‘plant perceptron’ by Scheres and van der Putten in their 2017 paper.

To avoid any chance of confusion / misinterpretation, we have removed the term “perceptron” and explained and rephrased it as the genetic network that it is.

Regarding the existence of an overarching genetic network that connects growth and environment: yes, we believe its existence is fully plausible but that it is important to spell it out to highlight what this network entails. It is important because it is the start point for addressing the question of the evolutionary origin of the genetic network that underpins the developmental plasticity of land plants. We pinpoint the homologous set of genes that might be acting in this process for more than 600 million years of streptophyte evolution.

Getting back to multicellularity, the discussions say, “a shared toolkit for multicellularity was present in the LCA of Zygnematophyceae, which comes to bear in filamentous genera but is still lingering as genetic potential in zygneumatophyte unicells”. This is certainly plausible, and I suspect it’s true, although I can’t say they proved it by not finding the requisite changes in protein domains.

>>>>AU: We think this is an important statement to make for the more general reader because it provides a contextualization and plausible explanation for our findings. Since it is in the discussion section and since we start the sentence with “A parsimonious explanation is...”, our feeling is that it is clear that it reflects our perspective—and that it is not the only possible explanation. We use this as an introduction to explain how we think our data relates to it (the sentences that follow).

Does this qualify for publication in a high-impact journal? Historically yes, because the data will be highly cited, even if the conclusions are tentative. As sequencing has gotten more routine, many authors feel compelled to hype their results. I understand and sympathize. But it’s borderline dishonest. If they would tone it down, I would be satisfied.

>>>>AU: Thank you for thinking deeply about our work, and for alerting us to the potential confusion over the term ‘perceptron’ — as noted above, we have removed it entirely from our manuscript. It was never our intention to hype/oversell our results. In revising our manuscript we have carefully considered the strength of our conclusions relative to our results, and have toned down aspects of our original text that might cause irritation. We hope that readers will now be better able to focus on the biological messages that we want to convey. As the reviewer noted, our work represents an important genomic resource for the community. Our sincere belief is that it also introduces and clarifies important concepts fundamental to understanding land plant evolution. These include co-expressed cohorts of genes whose links were previously thought to be specific to land plants, involved in sensing abiotic and biotic cues, as well as growth and development. The concertedness of their action might have emerged more than 600 million years ago and can now be scrutinized in light of this divergence. We believe that this reflects on the complexities when trying to decipher the molecular chassis for multicellularity from comparative genomic data—it is the expression (and ultimately regulation) that matters. Once again, we are grateful for the advice and feedback that R1 has given us — it has significantly improved our revised manuscript.

Reviewer #2:

Remarks to the Author:

I have read the latest version of the manuscript and the responses of the authors. I am now happy to recommend the manuscript for publication, with my apologies for submitting my review later than intended.

>>>>AU: We thank the reviewer for the kind words and for reviewing our work.

Decision Letter, second revision:

11th Jan 2024

Dear Dr. de Vries,

Thank you for submitting your revised manuscript "Genomes of multicellular algal sisters to land plants illuminate signaling network evolution" (NG-A62450R1). The reviewers find that the paper has improved in revision, and therefore we'll be happy in principle to publish it in Nature Genetics, pending minor revisions to comply with our editorial and formatting guidelines.

Sincerely,
Wei

Wei Li, PhD
Senior Editor
Nature Genetics
New York, NY 10004, USA
www.nature.com/ng

Final Decision Letter:

25th Mar 2024

Dear Dr. de Vries,

I am delighted to say that your manuscript "Genomes of multicellular algal sisters to land plants illuminate signaling network evolution" has been accepted for publication in an upcoming issue of Nature Genetics.

Your paper will be published online after we receive your corrections and will appear in print in the next available issue. You can find out your date of online publication by contacting the Nature Press Office (press@nature.com) after sending your e-proof corrections.

Please note that *Nature Genetics* is a Transformative Journal (TJ). Authors may publish their research with us through the traditional subscription access route or make their paper immediately open access through payment of an article-processing charge (APC). Authors will not be required to make a final decision about access to their article until it has been accepted. Find out more about Transformative Journals

Authors may need to take specific actions to achieve compliance with funder and institutional open access mandates. If your research is supported by a funder that requires immediate open access (e.g. according to Plan S principles) then you should select the gold OA route, and we will direct you to the compliant route where possible. For authors selecting the subscription publication route, the journal's standard licensing terms will need to be accepted, including [a href="https://www.nature.com/nature-portfolio/editorial-policies/self-archiving-and-license-to-publish"](https://www.nature.com/nature-portfolio/editorial-policies/self-archiving-and-license-to-publish). Those licensing terms will supersede any other terms that the author or any third party may assert apply to any version of the manuscript.

If you have not already done so, we invite you to upload the step-by-step protocols used in this manuscript to the Protocols Exchange, part of our on-line web resource, natureprotocols.com. If you complete the upload by the time you receive your manuscript proofs, we can insert links in your article that lead directly to the protocol details. Your protocol will be made freely available upon publication of your paper. By participating in natureprotocols.com, you are enabling researchers to more readily reproduce or adapt the methodology you use. [Natureprotocols.com](http://natureprotocols.com) is fully searchable, providing your protocols and paper with increased utility and visibility. Please submit your protocol to <https://protocolexchange.researchsquare.com/>. After entering your [nature.com](http://www.nature.com) username and password you will need to enter your manuscript number (NG-A62450R2). Further information can be found at <https://www.nature.com/nature-portfolio/editorial-policies/reporting-standards#protocols>

Sincerely,
Wei

Wei Li, PhD
Senior Editor
Nature Genetics
New York, NY 10004, USA
www.nature.com/ng